# Amortized Synthesis of Constrained Configurations Using a Differentiable Surrogate

**Xingyuan Sun**[1], **Tianju Xue**[2], **Szymon Rusinkiewicz**[1], **Ryan P. Adams**[1]
[1]Department of Computer Science    [2]Department of Civil and Environmental Engineering
Princeton University
{xs5, txue, smr, rpa}@princeton.edu

## Abstract

In design, fabrication, and control problems, we are often faced with the task of *synthesis*, in which we must generate an object or configuration that satisfies a set of constraints while maximizing one or more objective functions. The synthesis problem is typically characterized by a physical process in which many different realizations may achieve the goal. This many-to-one map presents challenges to the supervised learning of feed-forward synthesis, as the set of viable designs may have a complex structure. In addition, the non-differentiable nature of many physical simulations prevents efficient direct optimization. We address both of these problems with a two-stage neural network architecture that we may consider to be an autoencoder. We first learn the decoder: a differentiable surrogate that approximates the many-to-one physical realization process. We then learn the encoder, which maps from goal to design, while using the fixed decoder to evaluate the quality of the realization. We evaluate the approach on two case studies: extruder path planning in additive manufacturing and constrained soft robot inverse kinematics. We compare our approach to direct optimization of the design using the learned surrogate, and to supervised learning of the synthesis problem. We find that our approach produces higher quality solutions than supervised learning, while being competitive in quality with direct optimization, at a greatly reduced computational cost.

## 1 Introduction

One of the ambitions of artificial intelligence is to automate problems in design, fabrication, and control that demand efficient and accurate interfaces between machine learning algorithms and physical systems. Whether it is optimizing the topology of a mechanical structure or identifying the feasible paths for a manufacturing robot, we can often view these problems through the lens of *synthesis*. In a synthesis task, we seek configurations of a physical system that achieve certain desiderata while satisfying given constraints; *i.e.*, we must optimize a physically-realizable design.

In this work, *design* refers to the parametric space over which we have control and in which, *e.g.*, we optimize. A *realization* is the object that arises when the design is instantiated, while *goal* refers to its desired properties. For example, in fabrication, the design might be a set of assembly steps, the realization would be the resulting object, while the goal could be to match target dimensions while maximizing strength. Synthesis, then, refers to finding a design whose realization achieves the goal.

Synthesis problems are challenging for several reasons. The physical realization process may be costly and time-consuming, making evaluation of many designs difficult. Moreover, the realization process—or even a simulation of it—is generally not differentiable, rendering efficient gradient-based methods inapplicable. Finally, there may be a many-to-one map from the parametric space of feasible and equally-desirable designs to realizations; *i.e.*, there may be multiple ways to achieve the goal.

Surrogate modeling is widely used to address the first two challenges, though it can still be expensive because of the need for optimization, sampling, or search algorithms to find a feasible design. More seriously, the third challenge—lack of uniqueness—creates difficulties for naïve supervised learning

35th Conference on Neural Information Processing Systems (NeurIPS 2021).

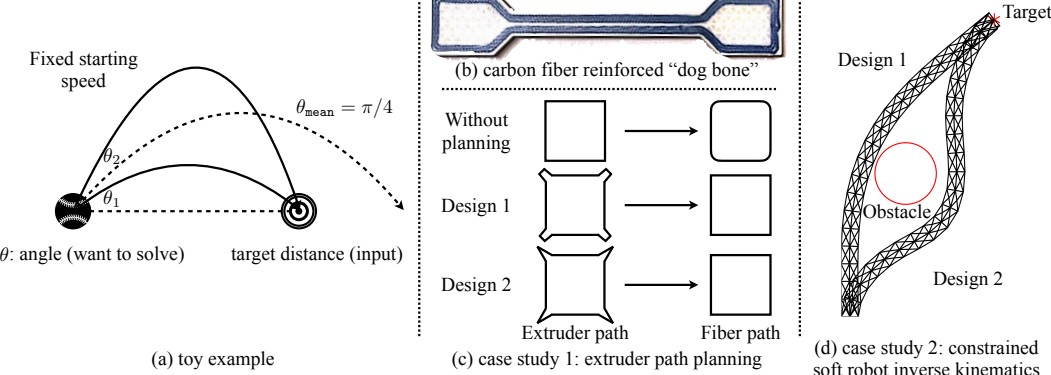

Figure 1: (a) For a fixed and large-enough starting speed, there exist exactly two angles such that the ball will hit the target, where the mean of these two angles is $\pi/4$. (b) Some 3D printers utilize fibers to reinforce the thermoplastic print. (c) For such printers, fiber is laid out along an extruder path but deforms into a smoothed version due to the fiber's high stiffness and low stretch. Our goal is to generate extruder paths that compensate for the smoothing, but multiple extruder paths can result in the same target shape, such as a square. (d) In soft robot inverse kinematics, we control the stretch ratios of both the left- and right-hand sides of a snake-like robot. Our goal is to reach the target while avoiding an obstacle but, as is illustrated, the solution is not unique – two different designs are shown.

approaches to synthesis. Specifically, consider generating many design/realization pairs, evaluating the constraints and objectives on the realizations, and attempting to learn a supervised map from goal back to design. When multiple designs lead to the same realization, or multiple realizations achieve the same goal, the supervised learner is penalized for producing designs that are valid but happen to not be the ones used to generate the data. Moreover, this approach may learn to produce an "average" design that is actually incorrect. Figure 1a shows a cartoon example: if the goal is to throw a ball to reach some target distance, there are two possible launch angles (designs) resulting in landing points (realizations) at the correct spot. Performing least-squares regression from distance to angle on the full set of distance/angle pairs, however, learns an average angle that does not satisfy the goal.

To address these challenges, we propose to use a two-stage neural network architecture that resembles an autoencoder. One stage (the decoder) acts as a differentiable surrogate capturing the many-to-one physical realization process. The other stage (the encoder) maps from a goal back to a design but, critically, it is trained end-to-end with a loss in the space of realizations that flows back through the decoder. Thus the encoder—our central object of interest for synthesis—is not constrained to match a *specific* design in a training dataset, but instead is tasked with finding *any* design that meets the desiderata of the realized output. The result is a neural network that performs *amortized* synthesis: it is trained once, and at run time produces a design that is approximately optimal, using only a feed-forward architecture. Note that our method is not an autoencoder, as the design is not a lower-dimensional representation of the goal, and the encoder and the decoder are trained in separate stages.

Our method places a number of requirements on the synthesis problem. First, to train the surrogate, we need data pairs of designs and realizations. Commonly, this would require us to generate designs, in which a substantial amount of designs are viable, and simulate them on a simulator. Second, given our current setting, the physical realization process needs to be deterministic. Third, to train the encoder, the synthesis problem should have a clear objective function, or at least we can quantify the objective. Finally, given our current feed-forward setting, we consider synthesis problems that need only one valid design, although we may further extend our method by using a generative encoder.

In this work, we demonstrate this two-stage approach on a pair of specific design tasks. The first case study is extruder path planning for a class of 3D printers (the Markforged Mark Two) that can reinforce polymer layers with discrete fibers (Figure 1b). Since the fibers are stiff, their shape is deformed after extrusion (Figure 1c, top row), and our task is to find an extruder path that results in a given fiber shape. As shown in Figure 1c, this problem has the many-to-one nature described above: for a small error tolerance on fiber path, there exist infinitely many extruder paths, which may even look very different. The second case study is constrained soft robot inverse kinematics. In this work, we use a simulation of a snake-like soft robot as in Xue et al. [97], in which we can control the stretch ratios of each individual segment on both sides of the robot. The robot has to reach

a target while avoiding an obstacle, and the locations of both are input goals. As before, there may be multiple solutions for a given goal (*i.e.*, locations of target and obstacle), as shown in Figure 1d.

For both case studies, we compare to two baseline algorithms. In direct-learning, a neural network for the synthesis problem (*i.e.*, from goal to design) is trained in a supervised manner on a set of designs. Since this effectively averages designs in the training dataset, as argued above, our method outperforms it significantly. The second baseline is direct-optimization, which uses a gradient-based method (BFGS) to optimize for each new design separately, given access to the trained differentiable surrogate for the realization process (decoder). Our method is competitive with this rough "performance upper bound" while using dramatically lower computational resources.

## 2   Related work

**Machine learning applications in synthesis problems.**   In synthesis tasks, the aim is to find a design solution such that its realization achieves one or more given goals. Usually, the solution is non-unique, and only one is needed. In molecule discovery, one would like to find a molecule which has some desired properties (*e.g.*, minimal side effects, efficacy, metabolic stability, growth inhibition) [20, 46, 71, 41, 83]. See Vamathevan et al. [90], Chen et al. [16] for surveys on machine/deep learning in drug discovery. Challenges of molecule discovery include discrete design space [32, 76] and limited data [47] due to difficulty in simulation. In materials synthesis, researchers seek materials with specific properties. See Bhuvaneswari et al. [7] for a review. Similarly, to obtain enough training data, researchers have put efforts into parsing and learning from scientific literature in natural language [52, 45]. In 3D shape generation, one wants to find a 3D shape that has some desired properties: properties like 2.5D sketches [95] that can be easily calculated and properties like functionality [34] that need to be human-labeled. In topology optimization, researchers aim to maximize the system's performance by optimizing the material layout given boundary conditions, constraints, external loads, etc [80, 5, 93]. Machine learning has been used to infer properties [75], find representations of designs [48, 99], and directly generate designs [55]. In program synthesis, the goal is to find programs that realize given intentions (*e.g.*, generating 3D shapes, answering visual questions) [35, 88, 98], where machine learning has been used to generate programs and execute programs. In this work, we propose an approach to learn a feed-forward neural network that can directly and efficiently produce a feasible solution to a synthesis problem that satisfies the requirements mentioned above.

**Surrogate/oracle-based synthesis.**   Surrogate/oracle-based synthesis uses an auxiliary model— a surrogate (or sometimes called oracle)—that can evaluate qualities of a design without time-consuming laboratory experiments, while still being reasonably accurate [58, 9]. Surrogate models can be physics-based or approximation-based [57] (*i.e.*, empirical), and there are different modeling techniques for approximation-based surrogates, including polynomial regression, radial basis functions, Gaussian processes, and neural networks [58, 39]. Bhosekar and Ierapetritou [6] provide a review of surrogate-based methods, and see Koziel and Leifsson [56] for a general review of surrogate models in engineering. There are two major approaches: optimization and sampling. The most common approach is surrogate-based optimization; see Forrester and Keane [29] for a survey. Some application examples include: optimizing the parameters of a CPU simulator [72]; solving partial differential equations in service of PDE-constrained optimization [97]; and optimizing stochastic non-differentiable simulators [79]. Researchers have also developed methods to deal with the challenge that inputs can be out-of-distribution for the oracle [27, 30, 89]. On the other hand, sampling-based methods have several advantages: the design space can be discrete and can generate multiple designs [9, 10, 24]. Researchers have successfully applied sampling methods to problems in chemistry [32] and biology [36, 51, 73]. In this work, we use surrogate-based optimization as one of our baseline algorithms, and our proposed method uses a surrogate during training to optimize for a feed-forward network for the design problem.

**Differentiable surrogate of losses in machine learning.**   Since some loss functions in machine learning are not differentiable (*e.g.*, IoU for rotated bounding boxes, 0-1 loss in classification), researchers have proposed to learn surrogates for them. Grabocka et al. [33] provided a formulation of surrogate loss learning and compared several learning mechanisms on some commonly used non-differentiable loss functions. Liu et al. [62] proposed a general pipeline to learn surrogate losses. Bao et al. [2], Hanneke et al. [40] provided theoretical analyses of surrogates for 0-1 loss in classification. Patel et al. [70], Nagendar et al. [64], Yuan et al. [100] explored the use of surrogate losses in various real-world tasks, including medical image classification, semantic segmentation,

and text detection and recognition. In this work, due to the non-uniqueness of design solutions, we further extend this idea from loss functions to more complex, physical realization processes.

**Robot motion planning.**    Robot motion planning [59] can also be viewed as a form of synthesis, and there are different types of approaches to it. One popular strategy is *optimization*. For example, researchers have used evolutionary algorithms (EA) [60, 12, 42], including variants of genetic algorithms (GA) [18, 17, 87, 11, 50, 53, 28] and covariance matrix adaptation evolution strategy (CMA-ES) [37]. Another example is constrained optimization [94, 85, 21, 23], which includes extensions like sequential quadratic programming (SQP) [15, 37] and the DIRECT algorithm [91]. Other examples of optimization applied to robot motion planning include reinforcement learning [81, 25] and teaching-learning-based optimization [82]. Another popular class of methods is *sampling*, including simulated annealing (SA) [4, 69, 101], probabilistic roadmaps (PRM) [49], rapidly exploring random trees (RRT) [14, 4, 13], *etc*. Finally, *search* methods, *e.g*., A* search [38], have been used to solve motion planning problems. These works usually model the robot's physics directly without using a surrogate model and solve the design using methods including optimization, sampling, and search, which can be time-consuming. Our method amortizes the cost of inference, and can be used to solve planning problems with more complex physics that cannot be directly modeled. In our first case study, we plan for a trajectory of the extruder, and our design space is not the speeds of the motors but rather the coordinates of points along the trajectory. In the second case study, rather than solving a dynamic planning problem, we solve a static planning task on a soft robot, with the stretch ratios of all the controllable segments of the robot as the design space.

**Path planning in 3D printing.**    Path planning is one of the most important problems in 3D printing, and people plan for different objectives: minimizing printing time, avoiding collision, faster planning, etc. Shembekar et al. [78] proposed a planning algorithm to build complex shapes with multiple curvatures and can avoid collisions. Ganganath et al. [31] minimized the printing time by modeling the task as a traveling salesman problem. Xiao et al. [96] speeded up path planning algorithms by introducing efficient topology reconstruction algorithms. Stragiotti [84] provided an optimization-based algorithm that minimizes compliance of a printed part. Asif [1] introduced a planning algorithm for continuous fiber and can generate a continuous deposition path. See Huang et al. [43] for a review of existing work in 3D printing path design. In this work, instead of the aforementioned objectives, we plan an extruder path to compensate for the deformation caused by the printing process. We demonstrate increased run-time efficiency by amortizing the cost of physical simulation of the printing process, learning a feed-forward network to output the extruder path.

## 3    Method

We formalize synthesis as a constrained optimization problem, denoting the set of allowed designs as $\Theta$ and the set of possible realizations as $\mathcal{U}$. There is a physical process that maps from design to realization that we denote as $U : \Theta \to \mathcal{U}$. Our goal may be a function of both the realization and the design, as designs may differ in, *e.g*., ease of manufacturing. Moreover, it may be appropriate to specify a parametric family of goals to accommodate related tasks, *e.g*., different target locations in inverse kinematics. The user expresses a ($g$-indexed) family of design goals via a cost function denoted $\mathcal{L}_g : \Theta \times \mathcal{U} \to \mathbb{R}$. The problem of interest is to optimize the cost function with respect to the design:

$$\min_{\boldsymbol{\theta} \in \Theta} \mathcal{L}_{\boldsymbol{g}}(\boldsymbol{\theta}, \boldsymbol{u}) \quad \text{s.t.} \quad U(\boldsymbol{\theta}) = \boldsymbol{u} \,. \tag{1}$$

We can view this problem as a generalization of PDE-constrained optimization problems, where we have allowed for broader types of realizations than PDE solutions. Revisiting the challenges of synthesis problems articulated earlier, $U$ may be expensive, non-differentiable, and non-injective, and $\mathcal{L}_g$ may not a have a unique minimum.

Recalling the toy problem of Figure 1a: $\boldsymbol{\theta}$ is the angle at which the ball is thrown, $U(\boldsymbol{\theta}) = \boldsymbol{u}$ is where it lands, the goal $\boldsymbol{g}$ is a desired distance, *i.e*., a target realization, and a (design-independent) cost function might be the squared difference between the desired and actual realizations: $\mathcal{L}_{\boldsymbol{g}}(\boldsymbol{\theta}, \boldsymbol{u}) = ||\boldsymbol{g} - \boldsymbol{u}||^2$.

Since the realization $\boldsymbol{u} = U(\boldsymbol{\theta})$ is unique for a specific design $\boldsymbol{\theta}$, we propose using a two-stage approach, which can be viewed as an autoencoder. We first learn a differentiable surrogate (decoder) $\hat{U}(\cdot)$ for the physical realization process from $\boldsymbol{\theta}$ to $\boldsymbol{u}$, and then learn an encoder $\phi(\cdot)$ from goal $\boldsymbol{g}$ to design $\boldsymbol{\theta}$, evaluating the design quality with the trained decoder. To build a dataset, we have to randomly sample designs and calculate realizations and goals from them, since the physical realization process $U(\cdot)$ is known, and the reverse direction (*i.e*., goal to design) is difficult as discussed before and is what we are trying to learn. Thus we first sample $D$ designs $\boldsymbol{\theta}_1, \cdots, \boldsymbol{\theta}_D$ and

build our dataset as $\mathcal{D} := \big\{ (\boldsymbol{\theta}_1, \boldsymbol{u}_1, \boldsymbol{g}_1), \cdots, (\boldsymbol{\theta}_D, \boldsymbol{u}_D, \boldsymbol{g}_D) \big\}$, where $\boldsymbol{u}_i := U(\boldsymbol{\theta}_i)$ and $\boldsymbol{g}_i$ is the goal calculated from the realization $\boldsymbol{u}_i$. The calculation of the goal $\boldsymbol{g}_i$ summarizes properties that we care about in $\boldsymbol{u}_i$, and this process depends on the synthesis problem itself and how the cost function $\mathcal{L}_.(\cdot, \cdot)$ is designed. We split $\mathcal{D}$ into $\mathcal{D}_{\text{train}}$, $\mathcal{D}_{\text{val}}$, and $\mathcal{D}_{\text{test}}$. We then train our surrogate $\hat{U}(\cdot)$ such that

$$\hat{U}^*(\cdot) := \arg\min_{\hat{U}(\cdot)} \mathbb{E}_{(\boldsymbol{\theta}, \boldsymbol{u}, \cdot) \sim \mathcal{D}_{\text{train}}} \big[ ||\hat{U}(\boldsymbol{\theta}) - \boldsymbol{u}||^2 \big], \tag{2}$$

so that $\hat{U}^*(\cdot)$ serves as a differentiable surrogate of the physical realization function $U(\cdot)$. We then use the trained decoder $\hat{U}^*(\cdot)$ to train an encoder $\phi(\cdot)$:

$$\phi^*(\cdot) := \arg\min_{\phi(\cdot)} \mathbb{E}_{(\cdot, \cdot, \boldsymbol{g}) \sim \mathcal{D}_{\text{train}}} \big[ \mathcal{L}_{\boldsymbol{g}} \big( \phi(\boldsymbol{g}), \hat{U}^*(\phi(\boldsymbol{g})) \big) \big]. \tag{3}$$

Note that although our method resembles an autoencoder, it is not an autoencoder, since we learn the decoder first and then the encoder rather than jointly. Besides, the design space is usually not a lower-dimensional representation of the goal space, and the cost function is usually not simply reconstruction loss.

## 4 Empirical evaluation approach

In this work, we use our method to solve two real problems as case studies: a task of optimizing the extruder path in additive manufacturing, and a task of actuating a soft robot in an inverse kinematics setting. While the details of the two differ, we evaluate them in the same way. In each case, we compare our algorithm with two baselines—`direct-learning` and `direct-optimization`—and evaluate our method in terms of relative quality and run time.

**`direct-learning`:**   One natural baseline is to directly learn a model from the goal $\boldsymbol{g}$ to the design $\boldsymbol{\theta}$. Thus we introduce the `direct-learning` model $\phi_{\text{dl}}(\cdot)$:

$$\phi_{\text{dl}}(\cdot) := \arg\min_{\phi(\cdot)} \mathbb{E}_{(\boldsymbol{\theta}, \cdot, \boldsymbol{g}) \sim \mathcal{D}_{\text{train}}} \big[ ||\boldsymbol{\theta} - \phi(\boldsymbol{g})||^2 + \mathcal{R}_{\text{dl}}(\phi(\boldsymbol{g})) \big]. \tag{4}$$

Note that, since there is no surrogate, we do not have access to the realization $\boldsymbol{u}$. We thus have to slightly adjust the cost function into a squared Euclidean distance part and a regularizer part $\mathcal{R}_{\text{dl}}(\cdot)$ (if there is one), the latter of which comes from the original cost function $\mathcal{L}_.(\cdot, \cdot)$. We expect our method to perform much better than `direct-learning`.

**`direct-optimization`:**   Since our surrogate is a neural network and therefore differentiable, another natural baseline is to directly optimize the design $\boldsymbol{\theta}$ with respect to the goal $\boldsymbol{g}$ by using a gradient-based optimizer (*e.g.*, BFGS), which provides us a rough performance "upper bound" on our method:

$$\phi_{\text{do}}(\boldsymbol{g}) := \arg\min_{\boldsymbol{\theta}} \mathcal{L}_{\boldsymbol{g}}(\boldsymbol{\theta}, \hat{U}^*(\boldsymbol{\theta})). \tag{5}$$

Note that AmorFEA [97] is state-of-the-art method for PDE-constrained optimization, which uses the same approach as the `direct-optimization` baseline. We expect our method to have performance close to `direct-optimization`, while running orders of magnitude faster.

To evaluate our method and the baselines, we iterate over all $\boldsymbol{g}$ from $\mathcal{D}_{\text{test}}$, and evaluate the quality of each tuple $\big( \phi(\boldsymbol{g}), U(\phi(\boldsymbol{g})), \boldsymbol{g} \big)$, where we use the physical realization function $U(\cdot)$ rather than the learned surrogate $\hat{U}^*(\cdot)$. We run every experiment 3 times with random initialization of neural networks. More details are provided in each case study and in the appendix.

## 5 Case study: extruder path planning

3D printing (a.k.a. additive manufacturing) is the process of creating a 3D object from a 3D model by successively adding layers of material. It has a wide variety of applications in areas including aerospace, automotive, healthcare, and architecture [77]. Popular 3D printers use thermoplastic polymers (PLA, ABS, nylon, etc.) as the printing material, but these have limited strength. To address this issue, some recent printers support using strong fibers to reinforce the composite. In this work, we explore a 3D printer (the Markforged Mark Two) capable of extruding discrete fibers (fiberglass, kevlar, or carbon fiber) along a controllable path. However, since the fibers are stiff and non-stretchable, the printed fiber path will be "smoothed" compared to the extruder path. As shown in Figure 1c, without path planning, the fiber path will be severely deformed. We seek to find a general method that, given any desired fiber path, plans an extruder path to compensate for the deformation caused by the printing process. To the best of our knowledge, there is no existing automated method for this task, so it is worthwhile to tackle it using machine learning.

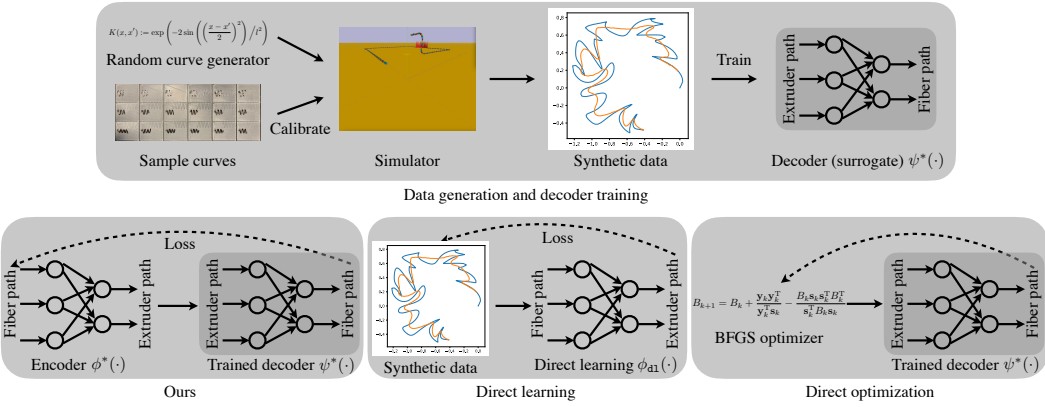

Figure 2: The pipeline for our method and two baselines for extruder path planning. We first generate data by building a simulator and calibrating it using a real printer, and we train the decoder and `direct-learning` using the synthetic dataset. We then train our method and build `direct-optimization` using the trained decoder.

## 5.1 Cost function

Following the notation in Section 3, we denote the target fiber path as $\boldsymbol{g} \in \mathbb{R}^{n \times 2}$: a path is represented as a series of $n$ points, and $n$ varies from path to path (at the scale of hundreds in our experiments). We denote the extruder path (the design) as $\boldsymbol{\theta} \in \mathbb{R}^{n \times 2}$ and its realization fiber path as $\boldsymbol{u} \in \mathbb{R}^{n \times 2}$. Our cost function $\mathcal{L}.(\cdot, \cdot)$ is defined as:

$$\mathcal{L}_{\boldsymbol{g}}(\boldsymbol{\theta}, \boldsymbol{u}) \coloneqq ||\boldsymbol{g} - \boldsymbol{u}||_2^2 + \lambda \cdot \mathcal{R}(\boldsymbol{\theta}), \tag{6}$$

where $\lambda$ is a hyper-parameter and $\mathcal{R}(\cdot)$ is a smoothing regularizer that calculates the sum of squared empirical second-order derivatives of the extruder path:

$$\mathcal{R}(\boldsymbol{\theta}) \coloneqq \sum_{i=2}^{n-1} \left( \left( \frac{\boldsymbol{\theta}_{i+1} - \boldsymbol{\theta}_i}{||\boldsymbol{\theta}_{i+1} - \boldsymbol{\theta}_i||_2} - \frac{\boldsymbol{\theta}_i - \boldsymbol{\theta}_{i-1}}{||\boldsymbol{\theta}_i - \boldsymbol{\theta}_{i-1}||_2} \right) \Big/ \left( \frac{||\boldsymbol{\theta}_{i+1} - \boldsymbol{\theta}_i||_2 + ||\boldsymbol{\theta}_i - \boldsymbol{\theta}_{i-1}||_2}{2} \right) \right)^2, \tag{7}$$

where $\boldsymbol{\theta}_i \in \mathbb{R}^2$ is the $i$-th row of $\boldsymbol{\theta}$.

## 5.2 Evaluation metric

The most intuitive way to evaluate the quality of extruder path $\boldsymbol{\theta}$ is to measure the distance between $\boldsymbol{g}$, the desired fiber path, and $\boldsymbol{u}$, the fiber path we get by printing $\boldsymbol{\theta}$. Note that it is likely the model under-estimates or over-estimates the deformation of fiber introduced by printing, so both cases can happen when we print with path $\boldsymbol{\theta}$: we run out of fiber before we finish $\boldsymbol{\theta}$, or there is still some fiber remaining in the nozzle after we finish $\boldsymbol{\theta}$ (the total length of fiber is fixed given a desired fiber path $\boldsymbol{g}$). In other words, the $\boldsymbol{g}_i$'s and $\boldsymbol{u}_i$'s might not be synchronized. Thus, directly measuring the distance between $\boldsymbol{g}_i$ and $\boldsymbol{u}_i$ does not necessarily reflect the difference between desired fiber path and the fiber path we get. Therefore, to better measure the distance between $\boldsymbol{g}$ and $\boldsymbol{u}$ during testing, we use Chamfer distance [26], which was first proposed by Barrow et al. [3] as an image matching technique, later developed as a commonly used (semi)metric to measure the difference between two sets, and has been shown to have a higher correlation with human judgment compared to intersection over union and earth mover's distance [86]:

$$d_{\mathrm{CD}}(\boldsymbol{g}, \boldsymbol{u}) \coloneqq \frac{1}{2} \left( \frac{1}{n} \sum_{i=1}^{n} \min_{j \in [1,n]} ||\boldsymbol{g}_i - \boldsymbol{u}_j||_2 + \frac{1}{n} \sum_{j=1}^{n} \min_{i \in [1,n]} ||\boldsymbol{g}_i - \boldsymbol{u}_j||_2 \right). \tag{8}$$

## 5.3 Implementation

The pipeline is shown in Figure 2. To generate the dataset for calibrating the decoder, we first use elliptical slice sampling [63] (New BSD License) to sample random extruder paths from a Gaussian process. We then use a physical simulator built using Bullet [22] (zlib License), calibrated to a real printer, to predict the realization (fiber path) for each extruder path. We generate 10,000 paths, split into 90% training, 5% validation, and 5% testing. For decoder, encoder, and `direct-learning`, we use an MLP with 5 hidden layers and ReLU as the activation function. The MLP takes 61 points as input and produces 1 point as output, and is applied in a "sliding window" fashion over the entire path (details in appendix). We train every model with a learning rate of $1 \times 10^{-3}$ for 10 epochs using PyTorch [68] and Adam optimizer [54]. For `direct-optimization`, we use the BFGS implementation in SciPy [92]. More implementation details are included in the appendix.

Table 1: Path-planning evaluation of the average Chamfer distance on the test set evaluated in simulation

| Regularizer weight | 0.1 | 0.3 | 0.6 | 1.0 | 1.5 |
|---|---|---|---|---|---|
| direct-learning | 0.0314±0.0008 | 0.0319±0.0016 | 0.0502±0.0072 | 0.1007±0.0292 | 0.1457±0.0216 |
| Ours | **0.0180±0.0004** | **0.0157±0.0003** | **0.0164±0.0004** | **0.0158±0.0002** | **0.0156±0.0002** |
| direct-optimization | | | 0.0155±0.0002 | | |

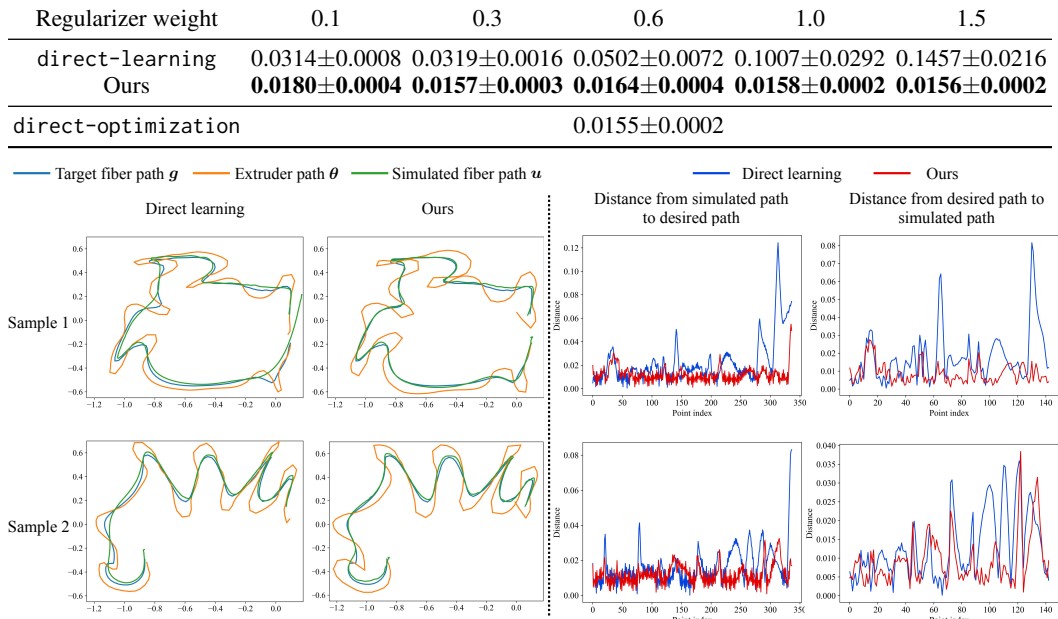

Figure 3: Path-planning evaluation of direct-learning *vs.* ours on the test set evaluated in simulation. We also visualize the Chamfer distance for each point on both the simulated fiber path and the desired fiber path.

## 5.4 Experiments

**Fiber path quality evaluated in simulation.** To quantitatively evaluate the effectiveness of our approach, the most straightforward way is to run its prediction on the simulator and see how close the simulated fiber path is to the input fiber path. We compare the performance of our approach (encoder), direct-learning, and direct-optimization on the test set of 500 paths, for different values of the regularization parameter $\lambda$. We report the average Chamfer distance (§ 5.2) with the standard error among 3 runs in Table 1. Note that direct-optimization runs very slowly, so we instead tune its regularizer weight on the first 40 test samples, select the best regularizer weight, and report its performance on the whole test set using the selected regularizer weight (more details in the appendix). The results demonstrate that our method significantly outperforms direct-learning, and the performance is comparable to direct-optimization.

Table 2: Path-planning evaluation of the average running time on the first 50 samples in the test set

| | Avg. time (s) |
|---|---|
| direct-learning | $7.96 \times 10^{-4}$ |
| Ours | $7.96 \times 10^{-4}$ |
| direct-optimization | $1.17 \times 10^{4}$ |

As a qualitative evaluation, Figure 3 shows the predictions of both direct-learning and our method on samples from the test set. We select the run with minimum average Chamfer distance over the test set for both direct-learning and our method, respectively. The results indicate that our method is better than direct-learning on handling details in the fiber path. We also visualize the Chamfer distance from each individual point on one path to the other, and observe that the distances for our method are generally lower than for direct-learning.

**Running time comparison.** To train needed neural networks, direct-optimization takes roughly 10 minutes, direct-learning takes roughly 1 hour, and our method takes roughly 5 days. Note that training costs are amortized, since we only need to train once. We then evaluate the inference time of the three algorithms on a server with two Intel(R) Xeon(R) E5-2699 v3 CPUs running at 2.30GHz. Since small neural networks generally run faster on the CPU, we run all of the tests solely on CPU. We run every algorithm on the first 50 paths in the test set and report the average inference time in Table 2. As we can see, both ours and direct-learning achieve a running time below 1 millisecond, while direct-optimization runs orders of magnitude slower.

**Fiber path quality evaluated on a real printer.** Lastly, we test our extruder path solutions on a real Markforged Mark Two 3D printer. We set the desired fiber path to a star, and print the star itself

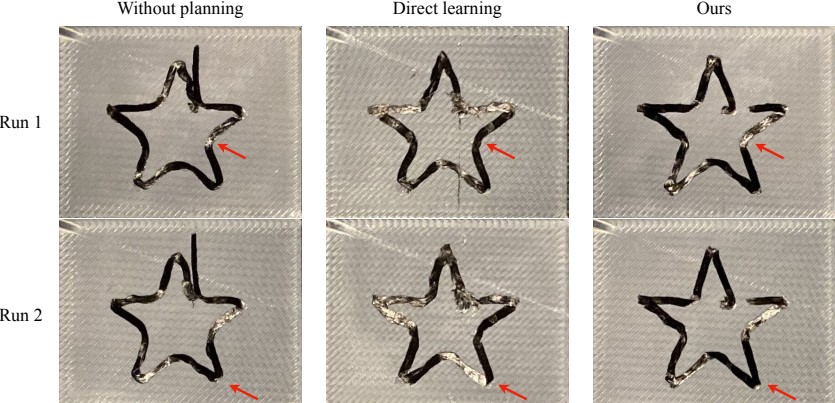

|  | Without planning | Direct learning | Ours |
|---|---|---|---|

Figure 4: Path-planning evaluation of "without planning" *vs.* `direct-learning` *vs.* ours on Markforged Mark Two, with a star as the desired fiber. For "without planning", we have the same extruder path for the 2 runs; for `direct-learning` and ours, the 2 runs are predictions from neural networks trained with different initializations.

(without planning) as well as solutions from both `direct-learning` and our method. We select regularizer weights based on Table 1, *i.e.*, 0.1 for `direct-learning`, 1.5 for our method, and we show two of the trained models. The results are visualized in Figure 4, confirming that our method successfully improves the quality of the printed fiber.

## 6 Case study: constrained soft robot inverse kinematics

Partial differential equations (PDEs) are a powerful tool for describing complex relationships between variables and have been used widely in areas including physics, engineering, and finance. In PDE-constrained optimization [8], the goal is to optimize a cost function such that the constraints can be written as PDEs, *i.e.*, the solutions are consistent with the relationships specified by the PDE. In the context of synthesis problems, we can consider the boundary conditions of the PDE as the design, and the solution of the PDE as the realization. Similar to other synthesis problems, different boundary conditions can result in the same PDE solution, and the cost function may not have a unique minimum. In the above situations, we propose to apply our method. We test our method on a specific PDE-constrained optimization problem—constrained soft robot inverse kinematics, which serves as a representative use case of our method in this large category of problems.

Soft robots made of elastic materials, have received significant recent attention because of their reduced potential harm when working with humans [74]. Researchers have explored a variety of applications, including surgical assistance [19], bio-mimicry [61], and human-robot interaction [67]. In this case study, as in Xue et al. [97], we use a snake-like soft robot with a fixed bottom, in which we can control the stretch ratios on both sides of the robot. As shown in Figure 1d, the objective is for the midpoint at the top of the robot to reach a target, while making sure that the robot does not collide with a fixed-size circular obstacle. The relationship between the robot's shape and the stretch ratios can be written as a PDE, and as shown in Figure 1d, there are different solutions to achieve the goal.

### 6.1 Cost Function

We adopt the soft robot from Xue et al. [97], which has an original height of 10 and an original width of 0.5, with its bottom is fixed. The goal vector $\boldsymbol{g}$ is in $\mathbb{R}^{2\times2}$, where $\boldsymbol{g}_1 \in \mathbb{R}^2$ indicates the target location and $\boldsymbol{g}_2 \in \mathbb{R}^2$ indicates the obstacle location. We denote the radius of the obstacle as $r$, which we set to $0.9$. The design (control) vector $\boldsymbol{\theta}$ is in $\mathbb{R}^n$ with $n = 40$, with $\theta_i \in \mathbb{R}$ indicating the stretch ratio of the $i$-th segment (*e.g.*, $\theta_i = 0.95$ indicates the $i$-th segment is contracted by 5%). The physical realization vector $\boldsymbol{u}$ is in $\mathbb{R}^{m\times2}$ with $m = 103$, where $\boldsymbol{u}_i \in \mathbb{R}^2$ indicates the location of the $i$-th vertex on the soft robot's mesh. The location of the top midpoint is denoted as $\boldsymbol{u}_{\mathtt{tm}} \in \mathbb{R}^2$. Implicitly, the relationship between $\boldsymbol{\theta}$ and $\boldsymbol{u}$ obeys a PDE that governs the deformation of the robot, as detailed in Xue et al. [97].

The cost function $\mathcal{L}_\cdot(\cdot, \cdot)$ is defined as

$$\mathcal{L}_{\boldsymbol{g}}(\boldsymbol{\theta}, \boldsymbol{u}) := \frac{1}{2}||\boldsymbol{g}_1 - \boldsymbol{u}_{\mathtt{tm}}||_2^2 + \lambda_1 \cdot \mathcal{B}(\boldsymbol{u}, \boldsymbol{g}_2) + \lambda_2 \cdot \mathcal{R}(\boldsymbol{\theta}). \tag{9}$$

The first term $||\boldsymbol{g}_1 - \boldsymbol{u}_{\mathtt{tm}}||_2^2$ is the squared Euclidean distance between the top midpoint of the robot and the target. The second term, weighted by a hyper-parameter $\lambda_1$ (which we fix at 0.5), enforces

Table 3: Soft-robot evaluation of the number of successful cases (over 1,000) on test set

| Regularizer weight | 0.03 | 0.05 | 0.07 | 0.09 |
|---|---|---|---|---|
| direct-learning | 907.7±3.1 | 918.3±3.4 | 910.7±3.4 | 912.3±3.8 |
| Ours | **986.3±0.5** | **975.0±3.9** | **981.7±5.0** | **984.7±4.5** |
| direct-optimization | 997.0±0.5 | 998.0±0.0 | 998.3±0.7 | 997.7±0.5 |

Table 4: Soft-robot evaluation of the average distance to the target on successful cases on test set

| Regularizer weight | 0.03 | 0.05 | 0.07 | 0.09 |
|---|---|---|---|---|
| direct-learning | 0.2171±0.0016 | 0.2200±0.0028 | 0.2236±0.0006 | 0.2179±0.0036 |
| Ours | **0.0657±0.0093** | **0.0464±0.0018** | **0.0599±0.0097** | **0.0691±0.0224** |
| direct-optimization | 0.0233±0.0003 | 0.0240±0.0004 | 0.0241±0.0004 | 0.0242±0.0003 |

the constraint via a barrier function [65, 66] for the obstacle $\mathcal{B}(\boldsymbol{u}, \boldsymbol{g}_2)$:

$$\mathcal{B}(\boldsymbol{u}, \boldsymbol{g}_2) := \frac{1}{m} \sum_{i=1}^{m} \left( \max(r + \Delta r - \|\boldsymbol{u}_i - \boldsymbol{g}_2\|_2, 0) \right)^2, \tag{10}$$

where $\Delta r$ is a hyper-parameter (which we fix at 0.1). A positive $\Delta r$ provides a penalty as well as nonzero gradients when the robot gets close to the obstacle. The last term contains another hyper-parameter $\lambda_2$, varied in our experiments, weighting a smooth regularization term $\mathcal{R}(\boldsymbol{\theta})$, with

$$\mathcal{R}(\boldsymbol{\theta}) := \frac{1}{n-4} \sum_{\substack{1 < i < n, i \neq n/2, \\ i \neq n/2+1}} \left( \frac{\theta_{i+1} - \theta_i}{2} - \frac{\theta_i - \theta_{i-1}}{2} \right)^2, \tag{11}$$

where $\theta_i$ for $i = 1, 2, \cdots, n/2$ corresponds to stretch ratios on the left-hand side of the robot, and $\theta_i$ for $i = n/2 + 1, \cdots, n$ corresponds to stretch ratios on the right-hand side of the robot. This regularizer prevents unphysical deformations with strong discontinuities.

## 6.2 Evaluation metric

Since there are two objectives—"reach" and "avoid"—in this task, we have two evaluation metrics. The first metric is the number of cases that successfully avoid the obstacle (*i.e.*, have all vertex positions outside the obstacle circle). The second metric is the average Euclidean distance of the robot's top midpoint to the target for successful cases.

## 6.3 Implementation

Using the finite element method [44] and the code from Xue et al. [97] (MIT license), we randomly generate 40,000 data samples, and split them into 90% training, 7.5% validation, 2.5% testing. For encoder, decoder, and direct-learning, we use an MLP with 3 hidden layers and ReLU activation. We train every model for 200 epochs with a learning rate of $1 \times 10^{-3}$ using PyTorch [68] and Adam optimizer [54]. For direct-optimization, we use the BFGS implementation in SciPy [92]. More implementation details are included in the appendix.

## 6.4 Experiments

**Design quality evaluation.** We experiment with different regularizer weights $\lambda_2$ for our method and the two baselines. During training, we randomly sample the location of the obstacle, and we ensure the robot never collides with the obstacle for direct-learning, since it does not have access to the realization vector and thus its loss function cannot contain the barrier function term for the obstacle (more details about direct-learning are in the appendix). For a fair comparison, during testing, we set the random seed to 0 such that for the same test sample, the obstacle will appear at the same location for all algorithms. The number of cases in which the robot successfully avoids the obstacle, with standard error for 3 runs, is shown in Table 3. The average Euclidean distance to the target for successful cases, with its standard error, is shown in Table 4. As the numbers show, our method is competitive to direct-optimization, and performs much better than direct-learning. Samples from the test set are shown in Figure 5 (for both algorithms, we select the best run with a regularizer weight of 0.5). We can see that our method collides less frequently while reaching the target more accurately than direct-learning.

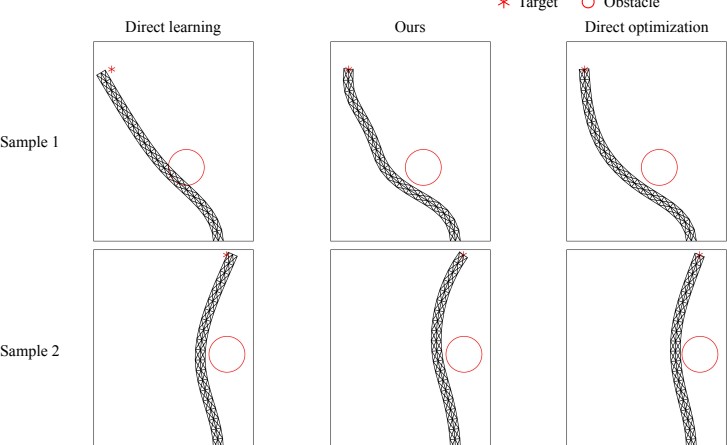

Figure 5: Soft-robot evaluation of `direct-learning` *vs.* ours *vs.* `direct-optimization` on examples from the test set. The direct learning baseline both violates the constraints (Sample 1) and fails to reach the target (Samples 1 and 2), while the run time of `direct-optimization` is 4000 times that of our method.

**Running time.** To train the neural networks, `direct-optimization` takes roughly 2 hours, `direct-learning` takes roughly 4 hours, and our method takes roughly 4.5 hours. Note that training costs are amortized, since we only need to train once. We then test all algorithms on a server with two Intel(R) Xeon(R) E5-2699 v3 CPUs running at 2.30GHz. Everything runs solely on the CPU, maximizing efficiency for the small neural networks we use. The results are shown in Table 5, showing that we successfully reduce the running time from over 1 second to less than 1 millisecond. Note that the soft robot is relatively small (40 control variables), and the time complexity of BFGS grows quadratically w.r.t. the number of parameters. Therefore, for more complex soft robots or PDE-constrained optimization problems with a larger number of variables, the running time advantage of our method can be of even greater importance.

Table 5: Soft-robot evaluation of the average running time on the test set

|  | Avg. time (s) |
|---|---|
| `direct-learning` | $3.12 \times 10^{-4}$ |
| Ours | $3.34 \times 10^{-4}$ |
| `direct-optimization` | $1.33 \times 10^{0}$ |

## 7 Discussion

In this work, we provided an amortized approach to synthesis problems in machine learning. To tackle the non-differentiability of physical system realizations, the huge computational cost of realization processes, and the non-uniqueness of the design solution, we designed a two-stage neural network architecture, where we first learn the decoder, a surrogate that approximates the realization processes, and then learn the encoder, which proposes a design for an input goal. We tested our approach on two case studies on fiber extruder path planning and constrained soft robot inverse kinematics, where we demonstrated that our method provides designs with much higher quality than supervised learning of the design problem, while being competitive in quality to and orders of magnitude faster than direct optimization of the design solution.

Although the experiments in both case studies show the effectiveness of our approach, we would like to mention some limitations of our method. First, to effectively learn a differentiable surrogate for the realization process, we need to be able to generate a substantial number of viable designs. We also need a simulator to calculate physical realizations of them, and the realization process has to be deterministic, although extensions might consider probability distributions over realizations. Also, to train the encoder, we need the objective ("goal") to be quantifiable. Our method provides the greatest gains if the realization is computationally expensive and/or non-differentiable, or if our encoder can exploit the non-uniqueness of designs to choose one good option where supervised learning would have learned a poor "average" solution. Additionally, due to the cost of neural network training, amortization is only a good idea when we need to solve one design problem many times with different goals, or we need fast inference. From a societal point of view, the primary negative consequence is the potential for replacing human labor in design. We view the present approach, however, as part of larger human-in-the-loop design processes in line with other software tools for modeling and fabrication.

## Acknowledgements

We would like to thank Geoffrey Roeder for helping set up the 3D printer, Amit Bermano, Jimmy Wu, and members of the Princeton Laboratory for Intelligent Probabilistic Systems for valuable discussions and feedback, as well as Markforged. This work is partially supported by the Princeton School of Engineering and Applied Science, as well as the U. S. National Science Foundation under grants #IIS-1815070 and #IIS-2007278.

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
