# A  Details about extruder path planning

## A.1  Methods

**Ours.**  As we discussed before in Section 3, we train a decoder (surrogate) using Equation 2 and an encoder using Equation 3, with the cost function defined in Equation 6. We take the trained encoder $\phi^*(\cdot)$ as our final model in use.

**direct-learning.**  As we discussed in Section 4, we train `direct-learning` using Equation 4, with a regularizer as defined in Equation 7. Note that this is equivalent to training to minimize the cost function $\mathcal{L}.(\cdot, \cdot)$ in Equation 6.

**direct-optimization.**  As described in Section 4, we build `direct-optimization` as in Equation 5, with a trained surrogate of the physical realization process. Here, to enforce that points $\boldsymbol{\theta}_i$ are evenly distributed along the extruder path, we use a slightly different cost function $\mathcal{L}_{\mathsf{do},\cdot}(\cdot, \cdot)$:

$$\mathcal{L}_{\mathsf{do},\boldsymbol{g}}(\boldsymbol{\theta}, \boldsymbol{u}) := d_{\mathsf{do}}(\boldsymbol{g}, \boldsymbol{u}) + \lambda_{\mathsf{do}} \cdot \mathcal{R}_{\mathsf{do}}(\boldsymbol{\theta}), \tag{12}$$

where we have a distance function $d_{\mathsf{do}}(\cdot, \cdot)$ and a smooth regularizer $\mathcal{R}_{\mathsf{do}}(\cdot)$. The smooth regularizer $\mathcal{R}_{\mathsf{do}}(\cdot)$ is derived from Equation 7 by requiring $\|\boldsymbol{\theta}_{i+1} - \boldsymbol{\theta}_i\|_2$ to be the same for all $i$:

$$\mathcal{R}_{\mathsf{do}}(\boldsymbol{\theta}) := \frac{1}{S_{\boldsymbol{\theta}}} \sum_{i=2}^{n-1} \left( \frac{\boldsymbol{\theta}_{i+1} - \boldsymbol{\theta}_i}{2} - \frac{\boldsymbol{\theta}_i - \boldsymbol{\theta}_{i-1}}{2} \right)^2, \tag{13}$$

where $S_{\boldsymbol{\theta}}$ is the length of the extruder path $\boldsymbol{\theta}$; this intrinsically enforces points in $\boldsymbol{\theta}$ to be evenly spaced. To measure the distance between $\boldsymbol{g}$ and $\boldsymbol{u}$, we first map them into two functions $\boldsymbol{f_g}(\cdot)$ and $\boldsymbol{f_u}(\cdot)$, such that $\boldsymbol{f_g}(s) \in \mathbb{R}^2$ is the location if we walk a distance $s$ along the path $\boldsymbol{g}$ (assuming the path is piecewise linear), and similarly for $\boldsymbol{f_u}(\cdot)$. Then the distance function is defined as

$$d_{\mathsf{do}}(\boldsymbol{g}, \boldsymbol{u}) := \int_0^1 \left\| \boldsymbol{f_g}(x \cdot S_{\boldsymbol{g}}) - \boldsymbol{f_u}(x \cdot S_{\boldsymbol{u}}) \right\|^2 \mathrm{d}x, \tag{14}$$

where $S_{\boldsymbol{g}}$ and $S_{\boldsymbol{u}}$ denote the lengths of paths $\boldsymbol{g}$ and $\boldsymbol{u}$, respectively.

## A.2  Data generation

**Random curve generation.**  To build the dataset, we first need to generate some random 2D curves, which can be used as extruder paths later. The curves should be smooth and non-intersecting. For each path, we take both the $x$ and $y$ to be Gaussian processes whose kernel function $K(\cdot, \cdot)$ is

$$K(x, x') := \exp\left( -\frac{\sin^2\big((x - x')/2\big)}{2\,l^2} \right), \tag{15}$$

with the two axes independent and $l = 0.1$. We use elliptical slice sampling [63] (New BSD License): for each path, we start from 1,000 points on a unit circle, and sample 1,000 times. To avoid intersections, we use a log-likelihood of $-\infty$ for a self-intersecting path, and a log-likelihood of 0 for a non-intersecting path. We generate 10,000 paths using this approach.

**Simulator.**  Since it is time-consuming to print every extruder path we generate in the last step on a real printer, we build a simulation system by using Bullet [22] to help us generate fiber paths. The simulator is also used in our evaluation. We calibrate the simulator on two materials—carbon fiber and Kevlar, respectively. To calibrate, we print paths shaped as sine functions with amplitudes ranging from 3.0 cm to 6.5 cm on the Markforged Mark Two printer (Figure 6). We then measure the amplitudes of the printed fiber paths, which will be lower than the amplitudes of the extruder paths because of smoothing, and fit lines to actual amplitude vs. extruder amplitude (Figure 7a). After that, we perform a grid search on the parameters of the simulator (such as stiffness and friction), run simulations with the sine functions as extruder paths, and collect data pairs of extruder path amplitudes and simulated fiber path amplitudes. We end up with calibrated simulators for carbon fiber (Figure 7b) and Kevlar (Figure 7c), by selecting the set of parameters for each having the minimum sum of squared distances between the fitted line from Figure 7a) and the collected simulation data points. Finally, we run the tuned simulators on the generated extruder paths. Though we have conducted experiments with both materials, due to space constraints, the experiments in the paper use data generated from the carbon fiber simulator. We visualize some extruder paths and simulated carbon fiber paths in Figure 8.

## A.3  Mapping from a path to another

We have to design a neural network that can take in an input path ($\mathbb{R}^{n \times 2}$) and output another path ($\mathbb{R}^{n \times 2}$), which can be used for encoder, decoder, and `direct-learning`. Both paths are sequences

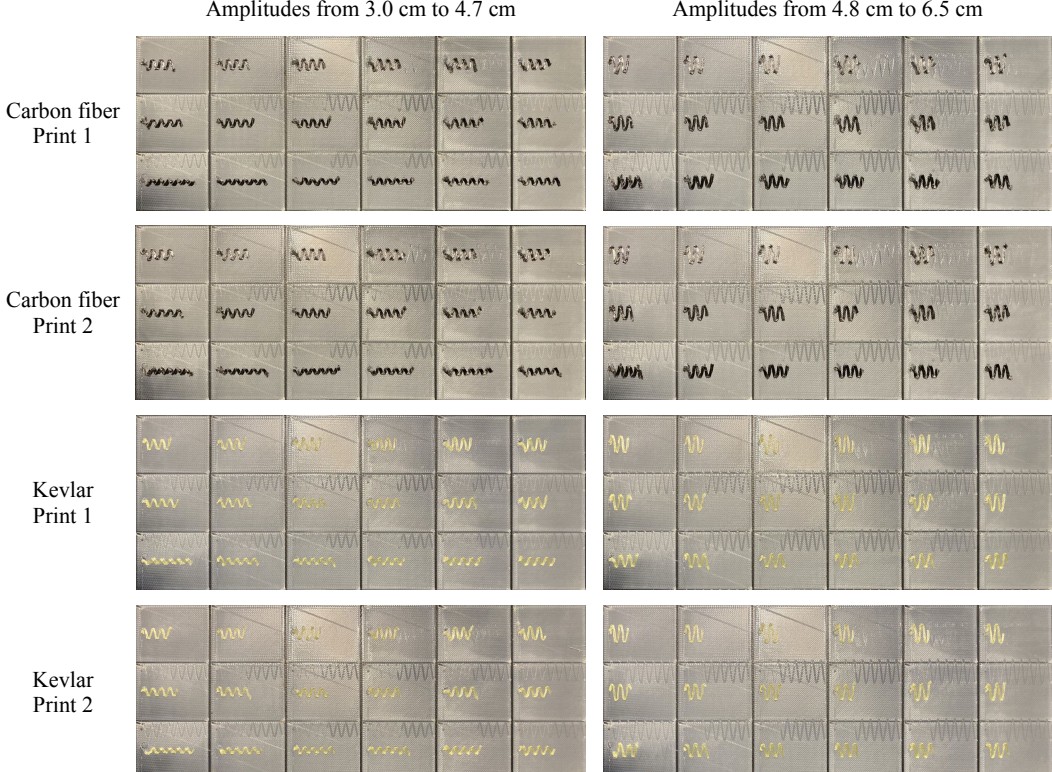

Figure 6: We print paths shaped as sine functions with amplitudes from 3.0 cm to 6.5 cm on the Markforged Mark Two printer, using carbon fiber and Kevlar, respectively. We print everything twice.

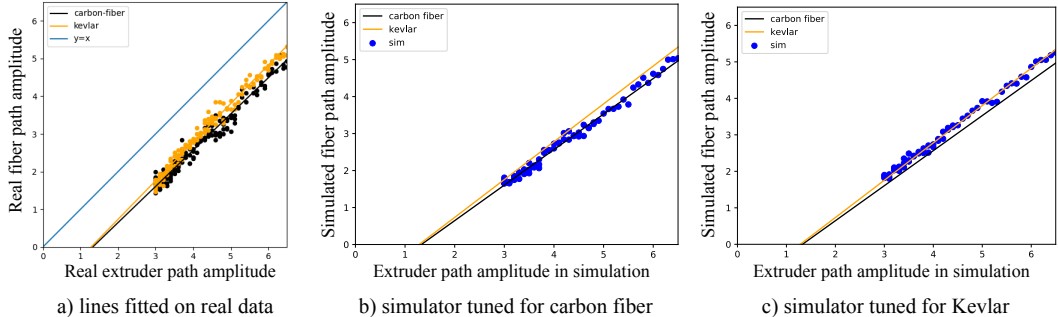

a) lines fitted on real data      b) simulator tuned for carbon fiber      c) simulator tuned for Kevlar

Figure 7: (a) We print paths shaped as sine functions with different amplitudes, collect amplitudes of the printed fiber paths, and fit lines through the data we collected. We experiment with two materials—carbon fiber and Kevlar, and the identity line is also visualized. (b) and (c) We select two sets of simulator hyper-parameters with their simulation results closest to the lines we get from the previous step for carbon fiber and Kevlar, respectively.

of $n$ 2D coordinates. For the purpose of illustration, we use decoder as an example here. Now the input is the extruder path $\boldsymbol{\theta}$, and the output is the resulting fiber path $\boldsymbol{u}$. Remember $\boldsymbol{\theta}_i \in \mathbb{R}^2$ is the $i$-th row of $\boldsymbol{\theta}$. Due to the intrinsic equivariant property of the problem, one natural idea is to have a neural network that takes in a certain number of points near $\boldsymbol{\theta}_i$ in $\boldsymbol{\theta}$ and outputs the corresponding point in $\boldsymbol{u}$ (*i.e.*, $\boldsymbol{u}_i$), and we iterate over every $i$, as a window sliding over $\boldsymbol{\theta}$. Note that we used the same neural network for all $i$'s.

We thus use a multilayer perceptron (MLP), which takes $2m + 1$ points (we set $m = 30$) and outputs one point. We take $\boldsymbol{\theta}_i$ as the starting point and resample $m$ points both forward and backward along the path $\boldsymbol{\theta}$. To be specific, as in § A.1, we first map $\boldsymbol{\theta}$ into a function $\boldsymbol{f}_{\boldsymbol{\theta}}(\cdot)$ such that $\boldsymbol{f}_{\boldsymbol{\theta}}(s) \in \mathbb{R}^2$ is the location if we start from $\boldsymbol{\theta}_1$ and walk a length of $s$ on the path. We further set $\boldsymbol{f}_{\boldsymbol{\theta}}(s) := \boldsymbol{f}_{\boldsymbol{\theta}}(0)$ for $s < 0$ and $\boldsymbol{f}_{\boldsymbol{\theta}}(s) := \boldsymbol{f}_{\boldsymbol{\theta}}(S_{\boldsymbol{\theta}})$ for $s > S_{\boldsymbol{\theta}}$, where $S_{\boldsymbol{\theta}}$ is the length of extruder path $\boldsymbol{\theta}$. We denote the distance of walking from $\boldsymbol{\theta}_1$ to $\boldsymbol{\theta}_i$ as $s_i$, *i.e.*, $\boldsymbol{f}_{\boldsymbol{\theta}}(s_i) = \boldsymbol{\theta}_i$. The input to the MLP is:

$$[\boldsymbol{f}_{-m}, \boldsymbol{f}_{-m+1}, \cdots, \boldsymbol{f}_0, \cdots, \boldsymbol{f}_m]^{\mathsf{T}}, \tag{16}$$

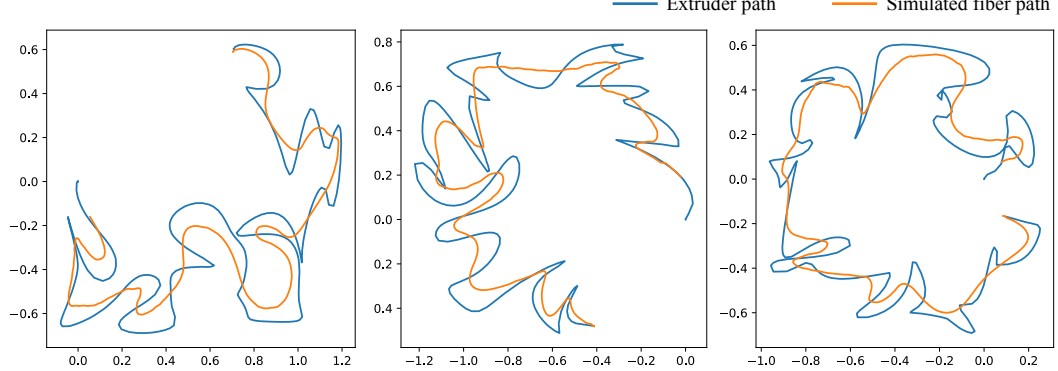

Figure 8: Samples from our dataset. We plot both the extruder paths and the simulated carbon fiber paths.

Table 6: The architecture of the MLP used in extruder path planning

| Type | Configurations |
|---|---|
| Fully connected | $4m$+2 to 500 |
| ReLU | N/A |
| Fully connected | 500 to 200 |
| ReLU | N/A |
| Fully connected | 200 to 100 |
| ReLU | N/A |
| Fully connected | 100 to 50 |
| ReLU | N/A |
| Fully connected | 50 to 25 |
| ReLU | N/A |
| Fully connected | 25 to 2 |

where

$$\boldsymbol{f}_i := \boldsymbol{f}_{\boldsymbol{\theta}}(s_i + i \cdot s_0) - \boldsymbol{f}_{\boldsymbol{\theta}}(s_i), \tag{17}$$

and $s_0 = 0.03$ is the step size. Since the problem is intrinsically translation-equivariant, we normalize every $\boldsymbol{f}_i$ by subtracting $\boldsymbol{f}_{\boldsymbol{\theta}}(s_i)$, as shown in the above equation.

### A.4  Hyper-parameters and neural network training

The architecture of the MLP is shown in Table 6, and we implement it in PyTorch [68]. We coarsely tuned the architecture, including the number of hidden layers (from 1 to 6) and the size of each hidden layer. We noticed that the accuracy is not largely affected by the architecture, as long as there is at least one hidden layer. We split the dataset into 90% training (9,000 paths), 5% validation (500 paths), and 5% testing (500 paths), and we use the Adam optimizer [54] with a learning rate of $1 \times 10^{-3}$, a learning rate exponential decay of 0.95 per epoch, and a batch size of 1 (path). We train every model—the decoder, the encoder, and `direct-learning`—for 10 epochs. We use our internal cluster with 7 servers with 14 Intel(R) Xeon(R) CPUs. For our method and `direct-learning`, we train them with different regularizer weights $\lambda = 0.1, 0.3, 0.6, 1.0$, and 1.5. For `direct-optimization`, we use the BFGS implementation in SciPy [92], with a gradient tolerance of $1 \times 10^{-7}$. Since its running time is extremely long, we tune its regularizer weight on the first 40 test samples (Table 7) and select the one with the best performance. To train the needed neural networks, it takes approximately 10 minutes for `direct-optimization`, approximately 1 hour for `direct-learning`, and approximately 5 days for our method. Note that we only need to train once so that the training costs are amortized.

## B  Details about constrained soft robot inverse kinematics

### B.1  Robot setting

We adopt the snake-like soft robot that was used in Xue et al. [97]. The robot has an original height of 10 and an original width of 0.5, and its bottom is fixed. We can control the stretch ratio of 40 segments (20 on the left-hand side and 20 on the right-hand side), as visualized in colors in Figure 9.

Table 7: Path-planning evaluation of `direct-optimization` of the average Chamfer distance on the first 40 samples in the test set evaluated in simulation

| Regularizer weight | 0.0001 | 0.0003 | 0.0006 | 0.001 |
|---|---|---|---|---|
| `direct-optimization` | 0.0171 | 0.0161 | **0.0153** | 0.0167 |

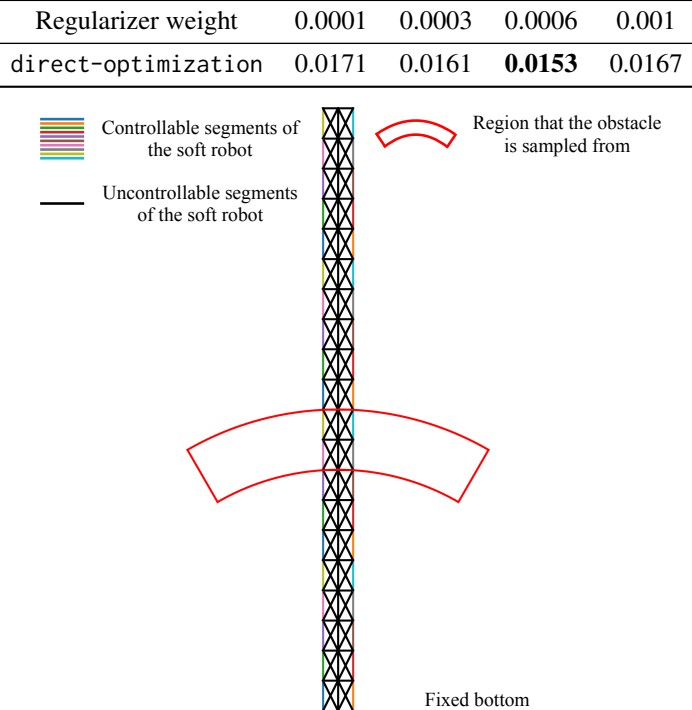

Figure 9: We visualize the soft robot with controllable segments in color, uncontrollable segments in black. The red region shows where we sample the center of the obstacle—a sector region with an angle of $60°$, inner and outer radiuses of 4 and 5, respectively.

The stretch ratios are restricted to be between 0.8 and 1.2. The physical realization of the robot consists of the locations of its 103 vertexes, as shown in Figure 9.

## B.2   Methods

**Ours.**   We follow Section 3 with the cost function defined as in Equation 9, and the obstacle location is randomly sampled.

**`direct-learning`.**   We follow Section 4 with $\mathcal{R}_{\mathrm{dl}}(\cdot)$ defined as in Equation 11. Note that since we do not have access to the physical realization $\boldsymbol{u}$, we cannot have a barrier function term for the obstacle. Thus, for `direct-learning`, we still randomly sample the obstacle location, but we guarantee during training, the obstacle does not collide with the robot in every specific training sample.

**`direct-optimization`.**   We follow Section 4 with the cost function defined as in Equation 9 and the obstacle location randomly sampled. Note that this baseline is similar to the approach used in Xue et al. [97]. The major difference is that we train the surrogate using supervised loss (as shown in Equation 2), and Xue et al. [97] trained their surrogate using a physically informed loss that minimizes the total potential energy.

## B.3   Data generation

We first randomly sample the design vector $\boldsymbol{\theta}$ with each dimension i.i.d. uniformly between 0.8 and 1.2. For each design vector $\boldsymbol{\theta}$, we solve the governing PDE with the finite element method [44] to obtain the corresponding physical realization $\boldsymbol{u}$ of the robot. Note that the obstacle location is randomly sampled during training and randomly sampled with a fixed random seed (we set to 0) during testing. The center of the obstacle is uniformly sampled from a sector region, as shown in Figure 9, with an angle of $60°$, an inner radius of 4, and an outer radius of 5. We altogether generate 40,000 data samples.

Table 8: The architecture of the MLP used in constrained soft robot inverse kinematics

| direct-learning | | Decoder | | Encoder | |
|---|---|---|---|---|---|
| Type | Configurations | Type | Configurations | Type | Configurations |
| Fully connected | 4 to 128 | Fully connected | 40 to 128 | Fully connected | 4 to 128 |
| ReLU | N/A | ReLU | N/A | ReLU | N/A |
| Fully connected | 128 to 256 | Fully connected | 128 to 256 | Fully connected | 128 to 256 |
| ReLU | N/A | ReLU | N/A | ReLU | N/A |
| Fully connected | 256 to 128 | Fully connected | 256 to 128 | Fully connected | 256 to 128 |
| ReLU | N/A | ReLU | N/A | ReLU | N/A |
| Fully connected | 128 to 40 | Fully connected | 128 to 206 | Fully connected | 128 to 40 |
| Sigmoid | N/A | / | / | Sigmoid | N/A |
| Linear map | $0.2(2x-1)$ | / | / | Linear map | $0.2(2x-1)$ |

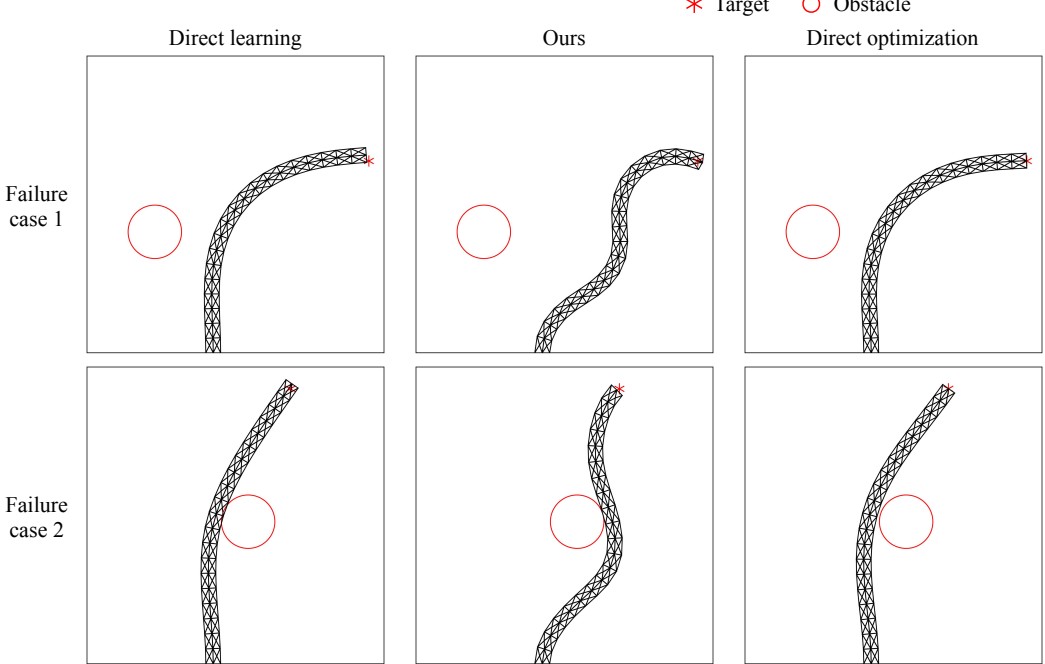

Figure 10: Failure cases of our method in test set on soft robot. In rare cases, our method might miss the target by a short distance (Samples 1 and 2) or touch the obstacle (Sample 2).

## B.4  Hyper-parameters and neural network training

We use MLP for all models (encoder, decoder, and direct-learning) with ReLU as the activation function and 3 hidden layers of sizes 128, 256, 128, respectively (Table 8). We coarsely tuned the architecture, including the number of hidden layers (from 1 to 4) and the size of each hidden layer. Similarly, we noticed that the accuracy is not largely affected by the architecture, as long as there is at least one hidden layer. Note that for the input and output of the neural network, we subtract 1 from all stretch ratios such that they are always between -0.2 and 0.2, and we use displacement of each vertex rather than its absolute location since displacement values are mostly centered around 0. In addition, to ensure that the encoder and direct-learning always output stretch ratios (minus one) between -0.2 and 0.2, we apply a sigmoid layer at the end of both the encoder and direct-learning, and linearly map the sigmoid output to be between -0.2 and 0.2 (as in Table 8). We use the same trick in direct-optimization to ensure the stretch ratios never fall out of range.

We implement all neural networks in PyTorch [68]. We split the dataset into 90% training (36,000 samples), 7.5% validation (3,000 samples), and 2.5% testing (1,000 samples), and we use the Adam optimizer [54] with a learning rate of $1 \times 10^{-3}$, a learning rate exponential decay of 0.98 per epoch, and a batch size of 8. We train every model—the decoder, the encoder, and direct-learning—for 200 epochs. For our method and all baselines, we experiment with different regularizer weights

Table 9: Ablation study: linear encoder vs. non-linear encoder for soft-robot evaluated on test set. All encoders are trained on non-linear decoders with a regularizer weight of 0.05

|  | #successful cases (over 1,000) | Avg. distance to target on successful cases |
| --- | --- | --- |
| Non-linear encoder | **975.0±3.9** | **0.0464±0.0018** |
| Linear encoder | 710.7±24.8 | 0.1324±0.0276 |

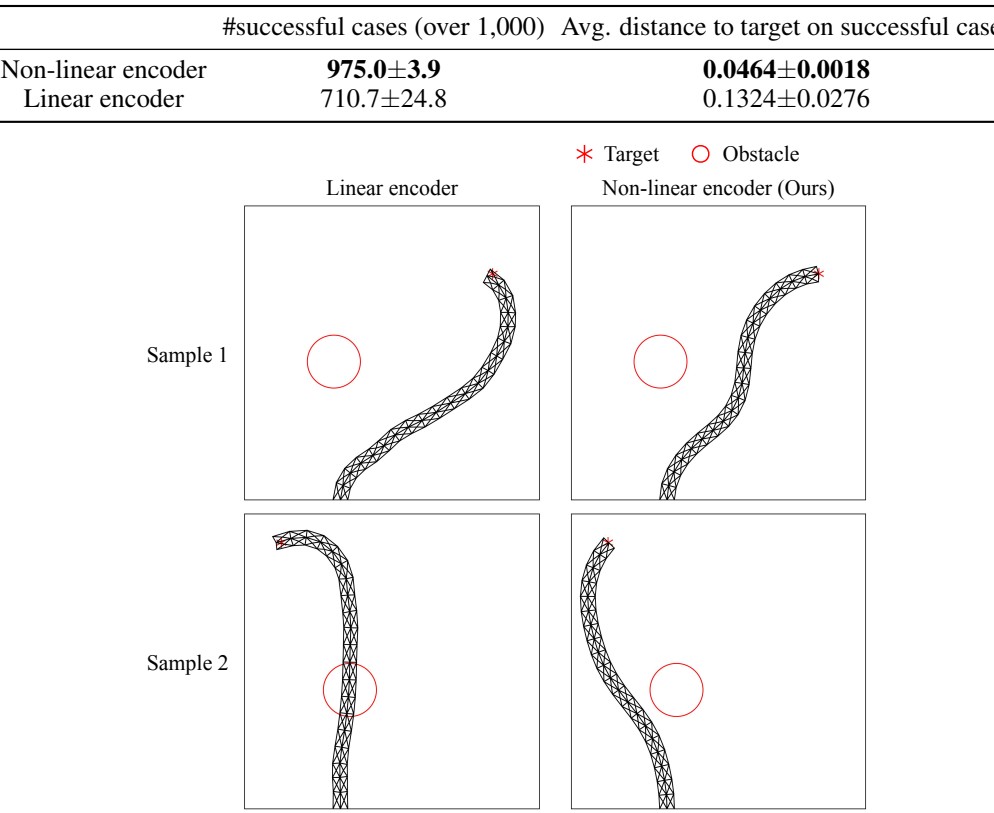

$\ast$ Target    ○ Obstacle

Figure 11: Soft-robot evaluation of linear encoder *vs.* non-linear encoder (ours) on examples from the test set. Linear encoder both violates the constraints (Sample 2) and misses the target (Samples 1 and 2).

$\lambda_2$ = 0.03, 0.05, 0.07, and 0.09. For `direct-optimization`, we use the BFGS implementation in SciPy [92], with a gradient tolerance of $1\times10^{-7}$. We use our internal cluster with 7 servers with 14 Intel(R) Xeon(R) CPUs. To train the needed neural networks, it takes approximately 2 hours for `direct-optimization`, approximately 4 hours for `direct-learning`, and approximately 4.5 hours for our method. Note that since we only need to train once, the training costs are amortized.

### B.5 Failure cases

Since our encoder and decoder are both neural networks, there might be some generalization errors. We show two failure cases of our method in test set in Figure 10. The design proposed by our method misses the target by a short distance in the first example, and both touches the obstacle and misses the target by a short distance in the second example.

### B.6 Ablation study: linear encoder

To demonstrate the non-linearity in our encoder is necessary, we train a linear encoder on the pre-trained non-linear decoder, following exactly the same training procedure mentioned in § B.4. We show the number of cases the robot successfully avoids the obstacle and the average Euclidean distance to the target for successful cases in Table 9. Linear encoder violates the obstacle constraints approximately 11.6 times as much as the non-linear encoder, and the average Euclidean distance for successful cases is approximately 2.9 times as much as the non-linear encoder. Two samples are shown in Figure 11. In both samples, the linear encoder misses the target by a short distance, and in the second sample, the linear encoder violates the obstacle constraint.