# OpenReview forum: "Amortized Synthesis of Constrained Configurations Using a Differentiable Surrogate"
_NeurIPS.cc/2021/Conference — NeurIPS 2021 Spotlight_

### Official Review · Reviewer_tJBF · 2021-07-09

**Rating:** 8
**Confidence:** 5

**Summary:**

The paper proposes a novel method for addressing synthesis problems, where they define a synthesis as an object or configuration that meets a set of constraints while optimizing one or more objectives. The authors motivate the problem with manufacturing settings, such as an additive printing path and extruding sequence, as well as soft robotic motion problem that can be described by a partial differential equation (PDE). The authors aims to address a couple of challenges:
* Provide a differentiable surrogate for non-differentiable simulations in synthesis problem. The claim and show that this can help with learning a differentiable model of the process that can help learn how to generate viable design via a synthesis process.
* Propose and develop a method that can solve many-to-one synthesis problems using a two-stage where one neural network based model learns how to generate designs and another model learns how to evaluate designs. Since both models are differentiable, the authors can train their framework end-to-end, which they claim provides accuracy improvements compared with directly learning a design to evaluation map and compute improvement compared to direct optimization of design to evaluation using method like BFGS.
* Provide a series of objective functions that can evaluate various synthesis design for a diverse set of use cases, which the authors then use to inform their two-stage approach.

The authors outline the objective function for an extruder path planning case and a constrained soft robotic motion task. In both cases they optimize the objective and perform the synthesis task using their approach as well as the direct learning (design -> evaluation) and direct optimization (BFGS) baseline. The authors claim, and the experimental supports, that the proposed two-stage method has a performance advantage of direct learning and a compute advantage, in inference, over direct optimization. The authors also discuss how even the training cost for their method is higher compared to direct optimization, the training cost gets amortized in inference as the two-stage does not require re-training/re-running like direct optimization.

**Limitations And Societal Impact:**

The authors briefly address limitations in the last section of the paper. As I mentioned in the previous section, it would be good to get a sense of scale of problems that authors think their method would be able to tackle. While the presented problems provide a good basis for proof-of-concept, what challenges do you anticipate in moving to larger and more challenging synthesis problems?
For the societal impact perspective, it would be good to include a discussion on the safety and reliability of methods like the ones the authors proposed would affect the over quality and process of synthesis processes, particularly in the manufacturing challenges discussed.

**Main Review:**

Nits:
* Checklist should start on a separate page after the references

Overall, I think the paper outlines a novel approach to a significant problem, provides thorough details on the method and relevant experiments, and has strong clarity in the writing and presentation. Please see the subsequent sections on suggestions on how to further improve the paper.

**Originality:** The paper tackles a new and relevant problem with a novel deep learning based method that involves end-to-end training of a differential surrogate model and a design generator. I think the approach is novel approach to synthesis problems the authors describe, which are a new set of challenges for deep learning methods.

**Quality:** The authors provide mostly thorough details on their method, formulation and experimental test cases. In order to improve the score, I would to get more clarity on the following questions:
* How does the "sliding window" work to generate an extruder path? Since the is sequential and time dependent, does the "encoder" produce the full directly or does it produce it sequentially? If is sequentially generated, how does the input change over time?
* What are the size of the design spaces for both cases? This would help get a sense of the scale of problems the paper tackled. It would also be useful to have a discussion on how the current approach could scale to larger and more complex challenges.
* Did you do an ablation of different loss functions? You discuss the importance of the Chamfer distance for the extrusion case, which is helpful, and I would be interested to learn more about how potentially naive loss function affect performance. I assume they would make things worse, but it would be helpful to get more detail on that.
* A discussion on future work/directions of the work would be nice.

**Clarity:** Overall, the paper is well written and presents relevant equations, formulations and diagrams to understand most of the relevant elements of the study. As mentioned above, to further improve on the clarify it would be helpful to get more detail on the input & output setups and relevant dimensions of the design cases, especially the outputs. There are more details on the inputs in the appendix, but the outputs are still somewhat unclear.

**Significance:** The paper addresses the application of a novel deep learning method to a non-traditional challenges that is significant.


**Time Spent Reviewing:**

2

---

> ### Author Response · Authors · 2021-08-10
> **Response to Reviewer tJBF**
>
> Thank you for your helpful review. Here are our detailed responses.
>
> 1. How the sliding window works. Thank you for asking this. The input is a sequence of n 2D coordinates, and we would like to generate the output sequence of the same length n, which can be produced parallelly (i.e., not sequentially). To get the i-th output, we have to send 2m+1 (m=30) points to the neural network, and we want them to be equally spaced. The (m+1)-th point (i.e., the one in the middle) in the input sequence is exactly the i-th input. We then start from the i-th input and walk both forward and backward along the input path (of n points), and we will get m points on both sides. By doing this resampling, the 2m+1 points can be used as the input to the neural network, so that the output is the i-th point in the output sequence. You may find more details in A.4 in the appendix, and please feel free to let us know if you have any further questions.
>
> 2. The size of the design space. For extruder path planning, the size of the design space depends on the length of the desired fiber path. For training, the design space has a dimension of 292. For testing on the real printer, we used a longer desired path, whose design space has a dimension of 582. For the soft robot task, the dimension of the design space is 40.
>
> 3. Generalization/future work. We will include more discussion about the generalization ability of our method, and future work. One promising direction is to investigate more complex neural network architectures of the encoder/decoder. For example, a generative encoder will be helpful if we need more than one design, and we can try RNN for designs with multiple parts and each part with continuous parameters, as mentioned by Reviewer PZD7.
>
> 4. Loss function. Thank you for pointing this out. We would like to clarify that we did not use Chamfer distance for training, but using it as an evaluation criterion. We use the squared loss for neural network training (as in equations 6 and 9 in the paper). For the evaluation, we use Chamfer distance since it is more similar to human judgment on the similarity between two shapes than other criteria.
>
> 5. Societal impact. We appreciate you for mentioning this. Our method provides much faster running time and at the same time without sacrificing much quality, which provides space for human-in-the-loop design, such that people only have to check the validity of the design or slightly adjust them. By doing this, we still need human labor and also provide better safety and reliability. We will include more discussion accordingly.

---

> > ### Comment · Reviewer_tJBF · 2021-08-31
> > **Thanks**
> >
> > Thank you for providing these additional details.

---

### Official Review · Reviewer_GztB · 2021-07-13

**Rating:** 7
**Confidence:** 2

**Summary:**

This paper considers two concrete problems in fabrication and design: path planning in 3D printing and soft robotic control. In both cases, there is trouble resulting from solutions that are not unique, in the sense that the inverse image from the codomain (solution space) is not injective. The paper proposes using a neural network architecture for feature space learning, which helps to collapse dimensionality in a way that allows the recovery of more robust (i.e. not the mean) solutions.

**Limitations And Societal Impact:**

The authors discuss the societal impact of automation of labor. As they point out, this type of algorithm is more of an IA proposition. There is no serious concern.

I would've liked to see further comparisons to different types of network architectures, even if only in an appendix. I'm guessing the network architecture is not very optimized. Since there was no benchmark presented where the author's method fails or even does somewhat poorly, it's hard to know where failure would occur. I always like to see more stress testing. Are neural networks even needed? What if we replace the encoder with a linear encoder? How well would that do? I'd just like to see the author's dig deeper into the model so that I can better understand why it works and where I can expect it to fail.

Finally, I should point out that fabrication is not my speciality. And I may have missed something.

**Main Review:**

This paper is well written and addresses an interesting problem. The authors showed two concrete design and fabrication problems where the use of surrogate objectives is useful for overcoming lack of uniqueness in synthesis problems. The results presented in table 3 seemed to convincingly show that the author's method outperforms naive baselines on a soft robotics problem.

The soft robotics problem, in particular, was a good setting. And I quiet enjoyed reading about the difficulties that arise in PDEs with non-unique solutions at the boundary. This is a very real world problem with an interesting machine learning solution, and I think it is exactly the type of work we should encourage at this venue.

Overall, the encoder-decoder architecture, such as it is, does not introduce anything new or interesting to the field. Indeed, the authors never make such claims. There are many alternative ways that the authors could go about this process, including adapting GAN-based and VAE-based approaches. Did the authors try any other approaches for their encoder-decoder architecture? Can you include a table with the dataset size and network parameters (optimizer, number of layers, etc)? I am a little confused about why the network took so long to train.

The data generation process also leaves something to be desired. Generating synthetic test data seems highly problem dependent. And the experiments in soft robotics in particular left me quite worried that the authors are just showing that a neural net can fit to a piecewise linear model. Or a simple piecewise convex model. With synthetic data, you always have to be quite careful that you're not simply investigating the ability of neural networks to approximate linear functions, etc. Although if Figure 4 is to be believed, it does seem like this works quite well in the real world.


The author's worked on interesting problems and developed a machine learning solution that significantly improves the final outcome. Surely, that's worth acknowledging.

**Time Spent Reviewing:**

2

---

> ### Author Response · Authors · 2021-08-10
> **Response to Reviewer GztB**
>
> Thank you for providing insightful feedback. Here are our detailed responses.
>
> 1. Encoder-decoder architecture. Thank you for pointing this out. We agree with you that GAN/VAE-based approaches are very interesting to try, especially when we need multiple designs rather than one. We currently focus on generating one feasible design, which MLP is capable of, and GAN/VAE-based approaches can be very promising followup work.
>
> 2. Long training time in extruder path planning. You may see the summary table below. As you already noticed, the training time of the encoder for path planning is relatively long (a few days), and the reason is we resample along the path to make sure points are equally spaced before sending them to the neural network, and this resampling process currently is not vectorized. This is an implementation issue specifically to the path planning task, which only affects the training time but not the inference time.
>
> ```
> Task                  Extruder path planning    Soft robot
> Dataset size          10,000 (paths)            40,000
> #MLP hidden layers    5                         3
> Optimizer             Adam                      Adam
> ```
>
> 3. Synthetic data issue. Thank you for bringing this up. We tested the performance of the linear encoder, and you can see the results below in “6. Linear encoder.”
>
> 4. Neural network architectures. Thank you for mentioning this. We will include further comparisons on different types of network architectures. We actually tuned the MLP for different numbers of hidden layers and hidden sizes. We first tried different numbers of hidden layers (from 1 to 6), and then briefly tuned the sizes of each hidden layer. As we show in “6. Linear encoder,” the nonlinearity from MLP is necessary for the two tasks we investigated.
>
> 5. Failure cases. Thank you for pointing this out. For example, for the soft robot task, as we can tell from table 4 in the paper, our method has a larger average distance to the target, although it might not be very easy to distinguish if we did not zoom in. We will provide zoomed-in figures to better illustrate the difference between the design of our method and “direct-optimization.”
>
> 6. Linear encoder. We agree this is an interesting ablation study that we can include in the paper. We train 3 linear encoders (i.e., single fully connected layer without activation function/nonlinearity) with different initializations. Note that for the surrogate, we still use MLP with 3 hidden layers such that it is still a reliable surrogate. As we can see in the table below, simply making the encoder linear significantly hurts the performance: about 11.5 times more frequently the robot collides with the obstacle, and the average distance to the target for successful cases is about 3 times as before. We therefore believe the nonlinearity provided by the neural networks is necessary for the soft robot task.
>
> ```
> Regularizer weight = 0.05                           Number of successful    Average distance to the target
>                                                     cases (over 1,000)      on successful cases
> Non-linear encoder trained on non-linear decoder    975.0±3.9               0.0464±0.0018
> Linear encoder trained on non-linear decoder        710.7±24.8              0.1324±0.0276
> ```

---

### Official Review · Reviewer_P3xV · 2021-07-16

**Rating:** 3
**Confidence:** 5

**Summary:**

The paper addresses the challenge of synthesis - where the goal is to find a configuration of a physical system that satisfies a set of constraints and optimizes one or more objective functions representing the desiderata. The synthesis challenge has three components - slow and costly realization process for evaluation of each design configuration, the physical realization or its software simulation being usually non-differentiable, and many-to-one mapping from the space of design configurations to the specification (constraints + objective function). Differentiable surrogates are used to address the first two challenges (to the extent possible for a realization process). The third challenge makes supervised learning difficult because the many-to-one mapping from configuration to realization to goal can penalize learned for good configurations/designs just because these were different from the training set. Adding more alternatives in the training set would also not help because the supervised learner would learn to predict average over the configuration space, which would semantically be not meaningful.

The solution proposed in this paper is similar to some other recent work in this space which use autoencoders - a decoder captures many-to-one mapping from design configuration to design goals, and another network that maps goal to a design.  As described in the limitations section, the paper's claim to novelty is not fully justified by their choice of benchmarks and baselines, and limited discussion of related work. Further, the reviewer is not convinced (for reasons described in the limitation section) that the presented approach would be practical even for physical design problems (ignoring the larger synthesis literature).

**Limitations And Societal Impact:**

- The baselines (direct-optimization, direct-learning) selected for comparison are rather crude and simplistic, and do not reflect the state-of-the-art in this area. The approach presented in this paper is described more generically as a general synthesis solution (and reviewer believes this to be the case too) and similar generic synthesis approaches have been developed with focus on slightly different application (proteins). For example, see Polykovskiy, Daniil, et al. "Entangled conditional adversarial autoencoder for de novo drug discovery." Molecular pharmaceutics 15.10 (2018): 4398-4405, Hawkins-Hooker, Alex, et al. "Generating functional protein variants with variational autoencoders." PLoS computational biology 17.2 (2021): e1008736. The reviewer is of the opinion that either the paper can be rewritten as one focussed on planning/robotics (with an explanation about why this domain is different from other synthesis problems) and the presented solution can be described as uniquely better suited to this domain, or synthesis techniques from chemistry/biochemistry need to be considered as baselines and the presented approach compared on some case-studies on that literature. These examples will also be more practically relevant challenges than the simple examples considered in the paper. In its present form, the paper's claim to novelty is not matched with its empirical analysis.

- More realistic benchmark problems in the robotics domain for evaluating the design approach would be to pick examples from state-of-the-art robotics papers on design: Campos, Thais, Samhita Marri, and Hadas Kress-Gazit. "Automated Synthesis of Modular Manipulators’ Structure and Control for Continuous Tasks around Obstacles." Robotics Science and Systems. 2020, Campos, Thais, et al. "Task-based design of ad-hoc modular manipulators." 2019 International Conference on Robotics and Automation (ICRA). IEEE, 2019.

-  The paper is also lacking discussion of related work to identify its incremental novelty over approaches developed there. Some relevant papers from ML venues include: Brookes et. al. "Conditioning by adaptive sampling for robust design." International conference on machine learning. PMLR, 2019. Fannjiang et. al. Autofocused oracles for model-based design. Advances in Neural Information Processing Systems, 33.

- Leaving aside the experimental evaluation and the core idea of using an autoencoder to go from goals to design space and back, the method proposed in the paper is rather straightforward and lacks any theoretical analysis of complexity or guarantees on the produced design. While a convincing experimental analysis to demonstrate practical utility would have been itself sufficient, the paper doesn't appear to have either major theoretical result or practical insight that could improve the state-of-the-art in synthesis.

- In real world, if we randomly sample designs, a vast majority would be actually invalid. Imagine putting together battery, speed control, propeller and motors to create a drone. In many cases, these combinations would produce current enough to burn the motor or torque high enough to tear the propeller. So, how do authors suggest building a dataset described in line 128-135. One reason why synthesis papers focus on biochemistry is this challenge of creating a dataset with physical designs. Path planning and kinematics are niche problems where each configuration (and configuration space is defined in a way to exclude impossible positions) leads to some measurable performance. These examples are not reflective of typical physical design problems.

- The paper adopts a rather simplistic approach to combined multiple objectives, which are common in design problems. The path planning example has "reach" and "avoid" as objectives, which are being weighed using hyperparameters. Typical physical design problems have several (10-100s) of optimization criteria and constraints, for e.g. a vehicle design might need one to look at track accuracy, thermal characteristics of motor, speed, etc. Combining them with hyperparameters would not be practically feasible (discovery of good values of these hyperparameters would be itself a hard problem).

**Main Review:**

+ The paper addresses a very important challenge of automatically finding configurations that meet design constraints and are optimum with respect to a given set of objectives.

+ The approach is demonstrated on a path planning (for a class of 3D printers) problem and a snake-like soft robotics inverse kinematics problem.

**Time Spent Reviewing:**

4

---

> ### Author Response · Authors · 2021-08-10
> **Response to Reviewer P3xV**
>
> Thank you for your thoughtful feedback. In summary, we would like to emphasize that:
> 1) the chemistry problem is different due to the lack of good simulators;
> 2) “direct-optimization” is state-of-the-art for PDE-Constrained Optimization;
> 3) our method resembles an autoencoder but is NOT an autoencoder;
> 4) our method is not aimed at higher quality but good quality with extremely fast inference;
> 5) there are a good variety of design tasks that our method can solve;
> 6) we believe our method to be useful as long as it can work on certain tasks (as we already showed two and mention two new examples in the following rebuttal);
> 7) compared to popular tasks like chemistry/biochemistry/rigid robots, we would like to bring methods and attention to less touched areas like 3D printing and soft robots so that they can all benefit from the ML community.
>
> Here are the detailed responses.
>
> 1. Baselines do not reflect state-of-the-art. Thank you for pointing out these papers. We will include a revised, more thorough version of related work. However, we would like to argue 1) the chemistry problem is different from the problems we are solving; 2) “direct-optimization” baseline is state-of-the-art for PDE-constrained optimization; 3) the approach of our work is different from the approaches in the papers you mentioned; our two-stage network is NOT an autoencoder.
>
> First, the chemistry problems you mentioned are different from the problems we are trying to solve in the paper. In small molecule chemistry, tools like density functional theory do allow a limited set of properties to be assessed, but nowhere near all of the functional properties researchers want to know for design.  Most biological activity properties of interest, as well as non-biological properties such as degradation time, must be assessed experimentally.  DFT and related methods are useful for energies and geometries of small molecules, but provide relatively limited information about reactivity.  Both papers you refer to used an autoencoder to reconstruct the chemical structures, and then used conditional approaches to generate new structures from the autoencoder’s latent space.  In one of the papers you mention, Polykovskiy et al. (2018), two of the three properties they use are logP and SA, which are not simulations, but simple functions called from RDKit that are known to be of limited predictive value.  Note also that for small molecules, high-quality retrosynthesis tools are not widely available, compared to the questions we would like to study in which fabrication is straightforward. Indeed, in chemistry the synthetic accessibility question is almost as important as the functional one; a common experience in the small molecule design space is to generate molecules that computational chemistry techniques assess as promising, but no one knows how to make.  Proteins are of course often easier to synthesize, but, due to the scale of the problem, simulation of protein function is even harder than performing computational chemistry for small molecules. (In the paper you mention, Hawkins-Hooker et al. (2021), luminescence is evaluated experimentally and not via simulation.) Taken together, we think these issues make it clear that the drug design problem is drastically different from the problems we study in this work.  For the synthesis problems we are investigating, there exist high-quality simulators, so that we can build surrogate models for them, and close the loop by building an encoder based on the surrogate.
>
> Second, “direct-optimization” is the method proposed in Xue. et al. “Amortized Finite Element Analysis for Fast PDE-Constrained Optimization”, ICML 2020, which “outperforms the traditional adjoint method on a per-iteration basis” for PDE-constrained optimization. In our work, we demonstrate that our method has comparable performance to it, while being much faster.
>
> Third, our method resembles an autoencoder but is NOT an autoencoder, and our method is different from the papers you mentioned. Compared to the two papers you mentioned, which finds the representation of chemicals using an autoencoder aiming at minimizing the reconstruction loss, in our work, the surrogate of the physical realization process is explicitly learned, given high-quality simulators are available, and we train an encoder explicitly minimizing the cost function. With the input and the output space of our two-stage neural network to be (potentially) different, our encoder and decoder not being trained simultaneously (when training the encoder, the decoder is frozen), and not minimizing the reconstruction loss, we argue our approach is different from using an autoencoder.
>
> 2. More realistic benchmark problems. Thank you for mentioning these papers. We would like to argue that 1) the papers you mentioned solve problems different from our paper 2) our approach does not need much special design for a specific task given a simulator/training data and a clear goal, and it is not aimed at providing designs with higher quality, but faster inference and reasonably good quality.
>
> First, in the soft robot case study, we deal with a continuum robot that, in theory, has infinitely many degrees of freedom, making the problem more difficult to solve, which is in the area of “soft robotics”. The mentioned papers solve problems with rigid robots composed of several one degree-of-freedom (DoF) modules.
>
> Second, our approach can be easily applied to design tasks given simulator/training data and a clear goal. We only need to train two neural networks on the data and slightly tune the network hyper-parameters, without needing specially designed rules for the specific task. Besides, the methods in both papers run in seconds/minutes, while the inference time of our approach is milliseconds.
>
> 3. Relevant papers from ML venues. Thank you for pointing these out, we will include more discussion on the difference between our method and other approaches. We want to emphasize that the papers you mentioned have a different focus compared to our work. In “Autofocused oracles for model-based design,” the main focus is on the gap between the surrogate and the real physical process, and in “Conditioning by adaptive sampling for robust design”, the main focus is on non-differentiable oracle model and discrete design space, while our work (comparing to them) is using surrogate to select one feasible design from infinitely many feasible solutions.
>
> 4. No theoretical result or practical insight. 1) As we mentioned above, “direct-optimization” is the state-of-the-art method in PDE-constrained optimization, and we provide comparable performance to it while being significantly faster, which will be very helpful as the dimension of design space becomes larger. 2) We already proved the effectiveness of our method in two case studies, and our method can potentially be applied to more tasks (e.g., fluid problems and computer graphics problems mentioned in the next paragraph).
>
> 5. Data generation can be difficult. Thank you for pointing this out. Configuration generation is beyond the scope of our paper, but we would like to argue there are still a great variety of real-world problems that we can solve given our current setting. Generating valid data within physical constraints can be challenging for problems like biochemistry, but for a vast range of other problems, this is not the case. For example, control problems in fluid mechanics (e.g., design external force with constraints on fluid speed) are widely studied, and building feasible datasets using high fidelity simulations is commonly accepted.  Similar problems have been studied within the field of computer graphics, ranging from art-directable physical simulation to computational fabrication.  Our method generalizes naturally to such problems.  More generally, domains in which the quality of a solution can be evaluated permit the use of techniques such as Markov Chain Monte Carlo (MCMC) to bias the sampling of data points (designs/realizations) to the region of high-quality solutions, making it feasible to generate large datasets efficiently.  Finally, even if the generated data fail to satisfy certain desiderata, our use of simulators permits these simulations to be run with no risk of physical damage, optimizing the ability of the surrogate to learn the behavior of the physical system near the constraint boundaries.
>
> 6. Tasks with a great many objectives. Thank you for mentioning this. We agree we did not state the contribution of our method clear enough and will revise accordingly. Our method cannot work on every design problem, and we believe there is no universal method that can work on every design problem. We believe our method to be useful as long as it can work well on a certain type of task: has high-quality simulators, clear objectives, data generation not too difficult. We already proved its effectiveness on 3D printing path planning and soft robots (also potentially control problems in fluid mechanics). These are less touched areas compared to popular tasks like chemistry/biochemistry/rigid robots. We would like to bring solutions and attention to these neglected tasks and they can benefit from the ML community.

---

> > ### Comment · Reviewer_P3xV · 2021-08-17
> > **Thanks to authors for detailed response. Reviewer's reply is interspersed.**
> >
> > The reviewer went over the response and the arguments do not appear convincing. The reply is interspersed. The incremental novelty of this work appears magnified due to the change of the usual context for such work (exemplified by citations from ML venue in the earlier review), and consequently the audience.  This change of context is not accompanied by significantly new identified challenges or algorithmic insights. The reviewer is in the minority with respect to score here, and would love to understand this paper better and change his/her perception of the work and the associated score, but would currently hold to the score in absence of some motivating arguments/data.
> >
> > Having said that, the reviewer is okay with having this review set aside if the consensus is otherwise. Given the current draft, the reviewer has to stick with the minority low score irrespective of the consensus.
> >
> > > First, the chemistry problems you mentioned are different from the problems we are trying to solve in the paper. In small molecule chemistry, tools like density functional theory do allow a limited set of properties to be assessed, but nowhere near all of the functional properties researchers want to know for design.
> >
> > The inverse kinematics problem and the simple planning problem present some of the simplest functional properties. There is some conflation of the overall goal of functional properties of design and what was actually accomplished in this paper.  Part of the paper appears to focus on the bigger goal of designing complex systems, but the actual technical content of the paper is limited to a much simpler setting and so, this challenge of dealing with "all of the functional properties" of design is not reflected in the technical body of the paper.  If the paper had actually developed and demonstrated an approach to design a new air-vehicle design or even a new wing design, the reviewer would have completely agreed with this claim.
> >
> > >  Indeed, in chemistry the synthetic accessibility question is almost as important as the functional one; a common experience in the small molecule design space is to generate molecules that computational chemistry techniques assess as promising, but no one knows how to make.
> >
> > Please note that the ML papers on this topic cited in the review are using synthesis in the sense of design similar to the literature in ML and formal methods on program synthesis and controller synthesis, and not the traditional biological/chemical synthesis. The reviewer was also pointing to that commonality and why those proposed approaches are applicable to the presented problems here and in reviewer's opinion worthy of experimental comparison with the presented approach.
> >
> > >  For the synthesis problems we are investigating, there exist high-quality simulators, so that we can build surrogate models for them, and close the loop by building an encoder based on the surrogate.
> >
> > This is exactly one of the reasons why this appears to be an easier version of the usually studied synthesis problem, where the available oracle is much more expensive to query. The reviewer fails to see why it is more challenging to build an encoder based surrogate for an easy-to-query oracle in the form of high-quality simulators, when compared to those methods where each query has to be judicious. The fact that surrogates are easy to optimize over, is not a consequence of some algorithmic insight, but rather a result of picking an easier class of synthesis problems.
> >
> > > Second, “direct-optimization” is the method proposed in Xue. et al. “Amortized Finite Element Analysis for Fast PDE-Constrained Optimization”, ICML 2020, which “outperforms the traditional adjoint method on a per-iteration basis” for PDE-constrained optimization. In our work, we demonstrate that our method has comparable performance to it, while being much faster.
> >
> > It appears the authors are convinced that comparison with the literature cited in the review will be unfair or is infeasible. The reviewer disagrees with the authors, and nothing in the rebuttal has provided any argument about why it is not possible. The reviewer fears that the paper has oversimplified the problem, demonstrated experimentally its value in a simpler setting against a weak baseline, and hence, this observation about experimental comparison with related work.
> >
> > >Third, our method resembles an autoencoder but is NOT an autoencoder, and our method is different from the papers you mentioned. Compared to the two papers you mentioned, which finds the representation of chemicals using an autoencoder aiming at minimizing the reconstruction loss, in our work, the surrogate of the physical realization process is explicitly learned, given high-quality simulators are available, and we train an encoder explicitly minimizing the cost function.
> >
> > Yes, the reviewer had made note of this distinction. The authors would perhaps agree that not training the encoder and decoder simultaneously (training encoder with decoder frozen) and minimizing cost function rather than reconstruction loss are minor tweaks and not a significantly deep insight into the right neural architecture for synthesis.
> >
> > > More realistic benchmark problems. Thank you for mentioning these papers. We would like to argue that 1) the papers you mentioned solve problems different from our paper 2) our approach does not need much special design for a specific task given a simulator/training data and a clear goal, and it is not aimed at providing designs with higher quality, but faster inference and reasonably good quality.
> >
> > Are the authors stating that their approach cannot be evaluated on these benchmark problems? Will such a statement be consistent with the generality claimed in the early part of the paper?
> >
> > > We want to emphasize that the papers you mentioned have a different focus compared to our work. In “Autofocused oracles for model-based design,” the main focus is on the gap between the surrogate and the real physical process, and in “Conditioning by adaptive sampling for robust design”, the main focus is on non-differentiable oracle model and discrete design space, while our work (comparing to them) is using surrogate to select one feasible design from infinitely many feasible solutions.
> >
> > Are authors stating that it is more challenging to work with differentiable oracle models compared to non-differentiable ones, specially if the differentiable models have been obtained by just restricting the problem to work with easy-to-evalute accurate simulators? Also, are the authors stating that the task of finding one of many solutions is an improvement over optimizing the solution?
> >
> > > Data generation can be difficult. Thank you for pointing this out. Configuration generation is beyond the scope of our paper, but we would like to argue there are still a great variety of real-world problems that we can solve given our current setting. Generating valid data within physical constraints can be challenging for problems like biochemistry, but for a vast range of other problems, this is not the case. For example, control problems in fluid mechanics (e.g., design external force with constraints on fluid speed) are widely studied, and building feasible datasets using high fidelity simulations is commonly accepted.
> >
> > The reviewer would beg to differ here. Take fluid mechanics (mentioned above by the authors) as the example - while a very simple setting where we have a static object composed of some primitive shapes, and a flowing liquid are easy to simulate in tools such as Fluent, typical design challenges require taking into account more intricate physics such as when does laminar flow separate from a surface and become turbulent? When does the presence of a small protrusion in the shape of an antenna cause this separation to happen much earlier than a smooth surface? This is the reason some literature the reviewer cited on synthesis against more realistic simulation oracles is critical. The paper is oversimplifying the problem (particularly when it comes to experimental evaluation) and thinking of this over-simplification as a novelty.
> >
> > > Tasks with a great many objectives. Thank you for mentioning this. We agree we did not state the contribution of our method clear enough and will revise accordingly.
> >
> > The review is thankful to authors for accepting the suggestion.
> >
> > > our method to be useful as long as it can work well on a certain type of task: has high-quality simulators, clear objectives, data generation not too difficult.
> >
> > When data generation is not difficult and high-quality simulators are available, it presents the easy end of the spectrum of synthesis/design problems for ML.
> >
> > The reviewer appreciates the authors for a detailed response and regrets that he/she is unable to revise the score in the absence of objective arguments or data-evidence. The reviewer is hopeful that the mentioned related work in the review will help provide authors a map of ML literature in the area and a better appreciation for the reviewer's concerns about this work's novelty and contribution to the area. The reviewer continues to believe that this work is not yet ready for publication in the form of the current draft.

---

> > > ### Author Response · Authors · 2021-08-26
> > > **Thank you for your detailed reply. Here are our responses**
> > >
> > > > The inverse kinematics problem and the simple planning problem present some of the simplest functional properties. There is some conflation of the overall goal of functional properties of design and what was actually accomplished in this paper. Part of the paper appears to focus on the bigger goal of designing complex systems, but the actual technical content of the paper is limited to a much simpler setting and so, this challenge of dealing with "all of the functional properties" of design is not reflected in the technical body of the paper. If the paper had actually developed and demonstrated an approach to design a new air-vehicle design or even a new wing design, the reviewer would have completely agreed with this claim.
> > >
> > > Thank you for this point. Our paper focuses on a faster method for design problems with some specific properties, and the inverse kinematics and path planning problems are two examples of them. For example, if we have a high-quality simulator for wing design, our method is likely to provide a shorter running time with reasonably good design quality. Besides, the problems that we are investigating are in a domain that is complex in its own way. We will make the scope of our method clearer in the paper.
> > >
> > > > This is exactly one of the reasons why this appears to be an easier version of the usually studied synthesis problem, where the available oracle is much more expensive to query. The reviewer fails to see why it is more challenging to build an encoder based surrogate for an easy-to-query oracle in the form of high-quality simulators, when compared to those methods where each query has to be judicious. The fact that surrogates are easy to optimize over, is not a consequence of some algorithmic insight, but rather a result of picking an easier class of synthesis problems.
> > >
> > > Thank you for your point. We argue that we provide a faster method for a specific type of design problem that you describe as “easy.” We do not agree they are easy, but there is already a surrogate-based optimization method that exists for them. Even if you describe them as “easy,” being easy is not a reason that we should stop exploring faster methods for them, and we do not see a reason that a faster method for “easy” problems is less valuable than a faster method for “difficult” problems. For example, as the paper we mentioned below [Kochkov, Dmitrii, et al. “Machine learning–accelerated computational fluid dynamics.” Proceedings of the National Academy of Sciences 118.21 (2021).], it is still meaningful to reduce the computational cost even if a high-quality simulator is available.
> > >
> > > > It appears the authors are convinced that comparison with the literature cited in the review will be unfair or is infeasible. The reviewer disagrees with the authors, and nothing in the rebuttal has provided any argument about why it is not possible. The reviewer fears that the paper has oversimplified the problem, demonstrated experimentally its value in a simpler setting against a weak baseline, and hence, this observation about experimental comparison with related work.
> > >
> > > We mentioned in the rebuttal that our method is for design problems with high-quality simulators. Thank you for this point, and we will make our scope clearer in the paper.
> > >
> > > > Yes, the reviewer had made note of this distinction. The authors would perhaps agree that not training the encoder and decoder simultaneously (training encoder with decoder frozen) and minimizing cost function rather than reconstruction loss are minor tweaks and not a significantly deep insight into the right neural architecture for synthesis.
> > >
> > > Thank you for your response. We do not agree these are minor tweaks. An autoencoder relies on the reconstruction loss to find a lower-dimensional representation of the inputs. In our case, the “latent space” in our “autoencoder” is directly supervised, and the “latent space” might even have a higher dimension than input/output, which violates these basic properties of an autoencoder. Besides, the “latent space” in our case is the design, which is obviously not a lower-dimensional representation of the input (i.e., the design target).
> > >
> > > > Are the authors stating that their approach cannot be evaluated on these benchmark problems? Will such a statement be consistent with the generality claimed in the early part of the paper?
> > >
> > > Our method needs a high-quality simulator such that we can train a differentiable surrogate on it. Thank you for mentioning these works, and we will make our scope clearer in the paper.
> > >
> > > > Are authors stating that it is more challenging to work with differentiable oracle models compared to non-differentiable ones, specially if the differentiable models have been obtained by just restricting the problem to work with easy-to-evalute accurate simulators? Also, are the authors stating that the task of finding one of many solutions is an improvement over optimizing the solution?
> > >
> > > Thank you for your reply. We are not stating which one is more challenging or not, but these papers are working on problems that are different from ours. We argue that our method runs faster than surrogate-based optimization while being reasonably accurate, as we demonstrated in the paper. As we wrote in the paper, for some design problems, there is a single minimum in the cost function, so that we can directly train a model from the target to the design. However, in some cases there are multiple good designs, so picking one solution from it becomes the new problem. Surrogate-based optimization is one solution, and we proved that our method provides (much) faster inference with reasonably good solution quality.
> > >
> > > > The reviewer would beg to differ here. Take fluid mechanics (mentioned above by the authors) as the example - while a very simple setting where we have a static object composed of some primitive shapes, and a flowing liquid are easy to simulate in tools such as Fluent, typical design challenges require taking into account more intricate physics such as when does laminar flow separate from a surface and become turbulent? When does the presence of a small protrusion in the shape of an antenna cause this separation to happen much earlier than a smooth surface? This is the reason some literature the reviewer cited on synthesis against more realistic simulation oracles is critical. The paper is oversimplifying the problem (particularly when it comes to experimental evaluation) and thinking of this over-simplification as a novelty.
> > >
> > > We thought that the original question raised by the reviewer was “in the real world, physically valid data generation is often impossible.” Therefore we used the fluids example to argue that there are (many) situations where physically reasonable data are generated just by numerical simulations and are used for training machine learning models. (See state-of-the-art [Kochkov, Dmitrii, et al. “Machine learning–accelerated computational fluid dynamics.” Proceedings of the National Academy of Sciences 118.21 (2021).]) This justifies the similar way we build our dataset. Simplifying the problem (e.g., without sophisticated experimental characterization) by numerical simulation is more of a common practice, not a novelty. In fact, we didn’t argue that as a novelty.
> > >
> > > > When data generation is not difficult and high-quality simulators are available, it presents the easy end of the spectrum of synthesis/design problems for ML.
> > >
> > > Thank you for your point. We do not agree with your view of the specific type of design problems we are solving. We believe all improvements on different types of problems should be encouraged. On the other hand, since surrogate-based optimization has provided high-quality design solutions for these problems, it is reasonable for us to provide a faster method that sacrifices a bit of quality.

---

### Official Review · Reviewer_PZD7 · 2021-07-17

**Rating:** 6
**Confidence:** 4

**Summary:**

The core concern of this paper is the synthesis problem of choosing design parameters in a complex engineering design process mapping a high level goal to a high dim description of how the process achieves it. This is studied using the examples of 3D printing path planning and trajectory generation for a soft robotic arm.

The core contribution is to set up this problem in terms of two stages - mapping goals to a space of designs, and then an evaluation of designs in terms of a cost function. This breaks down one of the difficulties identified - the ill-posedness of the problem due to multi-valued maps. So, the authors set up an autoencoder architecture with an encoder that conjectures designs based on goals, and a surrogate model to evaluate the designs.

**Ethical Concerns:**

-

**Limitations And Societal Impact:**

This paper is a methods contribution which is likely to be used in systems such as 3D printers and robots. I believe the authors have taken due care in setting up and disclosing their software use.

**Main Review:**

This is a clearly written paper on a timely topic.

The core contribution is the formulation factoring the evaluation of the design from the mapping of goals to designs, which breaks down the problem of many-valued mapping. Both parts are defined in terms of a differentiable module, so that the training can be achieved jointly. The evaluation methodology addresses what I would have expected to see - setting up baselines to show the extent to which each piece is relevant. So, one baseline asks how much is achieved by just mapping from goals to designs from historical data, while another asks how much is achieved by optimizing designs given the trained surrogate or decoder.

I have only a few points of critique that I would have liked to have seen expanded in this paper. This is based on the acknowledgement that the technically arguments seem essentially sound to me, so my comments are mainly on formulation.

(1) Building on the notion of multiple ways for a design to be good, many engineering designs have path-dependent constraints. For instance, while moving fast there may be a different cost to lateral deviations than when moving slowly. This is a combination of dynamics (something not considered explicitly in the examples) and other forms of heterscedasticity or path dependence. To the extent that the surrogate and the decoder are fairly simple MLPs, it is unclear if such phenomena can be handled. I think this is an important consideration, especially if the use cases include robotics and cyber-physical domains involving control.

(2) While the introduction talks about considerations beyond the basic path, e.g., "ease of manufacturing" mentioned in sec 3, they do not feature later on. So, for instance, the soft body reaching problem could have been solved to some extent by multi-rigid body assumptions without seriously needing to acknowledge the infinite-dimensional space. On the other hand, there are design problems involving fundamentally different designs with discrete and continuous choices, e.g., multiple body morphologies involving soft components each (just for argument's sake). It is not clear if the proposed method would apply as described in these settings - calling for a clarification of generality and scope.

(3) In the same vein, the design objectives here do admit a clear quantitative description always, e.g., distance from a desired path and reach error. There are many design problems where the objective is qualitative (e.g., any design from a topologically defined class will suffice). I expect that this might be easier to encode in this architecture but an example of this kind would have been helpful.

In summary, this is a clearly written paper whose methodology is sound. It would be stronger if the formulation's assumptions and limits are more clearly delineated.



**Time Spent Reviewing:**

1.5

---

> ### Author Response · Authors · 2021-08-10
> **Response to Reviewer PZD7**
>
> Thank you for your constructive review. Here are our detailed responses.
>
> 1. Path-dependent constraints/multiple ways for a design to be good. Thank you for pointing this out. We agree that the quality of a design might not only depend on the final physical realization but also the design itself, so our cost function also takes the design as an input. For example, in 3D printing path planning, given the same printing result, we might want to prefer a design with a shorter printing time. Although we use a constant printing speed in our experiment, we can generalize our method to varying printing speed: we can change the inputs of our encoder/decoder to be equally spaced on time (i.e., the time difference between input coordinate i and input coordinate i + 1 to be identical). To do this, we will need to re-generate the dataset with paths with the extruder moving at a varying speed. Finally, the penalization of printing time can be easily added as a term in the cost function. For more complex tasks, we agree with you that more complex network architectures might be needed. For example, to generate polygon mesh, a graph neural network might be needed.
>
> 2. Considerations beyond the basic path. Thank you for bringing this up. We agree with you that our method indeed can be generalized to problems involving rigid robots (discrete choices).
> The proposed methodology remains the same, and we only need to modify the representations of data space as well as the architecture of the networks. On the other hand, we agree that in some cases, the design can be a combination of discrete and continuous choices, and we need more complex neural network architectures in this case. For example, for a design with multiple parts and each part with continuous parameters, our encoder/decoder can be recurrent neural networks (or other fancier models), which can consume/generate a sequence of data.
>
> 3. Qualitative design objective. We agree that there are some objectives that are not easy to quantify. In this case, we incorporate its quantification process into the physical realization process/decoder/surrogate. For example, in 3D printing path planning, let’s assume we want the extruder path to be visually appealing. To do this, we can collect people’s scores on every extruder path in our dataset using crowdsourcing and train a neural network on it, predicting an “appealing” score.

---

> > ### Comment · Reviewer_PZD7 · 2021-08-27
> > **Thanks**
> >
> > I thank the reviewers for their response.
> >
> > Re. (1), I realize that one could indeed expand the dataset. However, the point is that properly representing all possible velocity profiles and other such higher order requirements would significantly expand the domain for search, hence the amount of data/variations needed (speaking loosely, you will have the product of infinite dimensional spaces, unless one makes restrictive assumptions). I accept that there may be modified versions of these architectures that could address such issues, but this is one example from many more (including the detailed discussion the authors have had with the other reviewers) of the need to temper claims by better delineating the scope of what is covered in the synthesis.
> >
> > Re. (2), the point I want to emphasize is that while the example has a continuum robot, the task does not demand that the infinite dimensionality be meaningfully considered. So, as just an illustrative example, planning with a much lower dimensional approximation to the arm and then enforcing the best estimate on the continuum system would suffice. In this sense, the task is 'simple' and does not demonstrate all of the complexities of soft robotics (as a roboticist would understand the issues). So, once again, clarifying the scope would improve the paper.
> >
> > That said, I had already expressed support for the paper in my original score, which I leave unchanged.

---

### Decision · Program_Chairs · 2021-09-27

**Decision:**

Accept (Spotlight)

**Comment:**

Thank you for your submission to NeurIPS.  Even after discussion, there is some substantial disagreement on this paper, so I will need to express an opinion that is not a unanimous one.  But with this caveat, I will state that I ultimately come down very much on the "positive" side of the disagreement, and I am recommending the paper be accepted to NeurIPS as a spotlight.

The rationale for this is rather simple: although currently confined to rather simple domains, the paper makes a compelling case that introducing a differentiable surrogate into synthesis problems, in a disciplined and generic fashion, can provide substantial gains over direct learning on the system of interest.  The authors do an extremely good job presenting detailed (and real, even if they are simple) evaluations of this strategy.  Thus, in total the authors present a compelling (_in_ its simplicity, rather than this being a flaw) approach to constrained synthesis, and show that across multiple realistic domains it outperforms the most common baseline.  Those are all points of a strong paper.

Now let me address the negative reviewer's concerns, which in my mind boils down to the concern that 1) the domains evaluated here are not particularly challenging and 2) don't reflect the vast amount of work that is already done in incorporating neural networks into synthesis problems.   I think it would of course be good to mention some of these connections (in the discussion, not in direct comparison) in the final draft of the paper.  But it is also the case that the landscape of integrating neural networks into specific physical synthesis domains is _vast_, and attempting to consider the scope in which all these approaches have been applied previously wouldn't be feasible.  Yes, for any given domain there are likely multiple NN-based solutions to synthesis problems, but attempting to consider all of these, in my opinion, would make anything but extremely domain-specific papers impossible.  The current paper makes no promises of extending the state of the art in a given practical domain (say the chemistry domains mentioned by the reviewer), but does offer evidence of a generic and compelling approach, and I believe these should be welcomed even if they are not (yet) as in-depth within a given domain as would be ideal in the longer term.  I think this is likely to be a fundamental disagreement as to the "right" focuses of papers like these, so while I don't expect to resolve this dilemma (and I certainly understand the reviewer's opinion), I still lean strongly towards accepting the paper.